# The Time-Energy Model: Selective Time-Series Forecasting Using Energy-Based Models

**Jonas Brusokas**                                                                 *jonasb@cs.aau.dk*
*Department of Computer Science*
*Aalborg University*

**Seshu Tirupathi**                                                                 *seshutir@ie.ibm.com*
*IBM Research, Ireland*

**Dalin Zhang**                                                                     *dalinz@cs.aau.dk*
*Department of Computer Science*
*Aalborg University*

**Torben Bach Pedersen**                                                            *tbp@cs.aau.dk*
*Department of Computer Science*
*Aalborg University*

**Reviewed on OpenReview:** *https://openreview.net/forum?id=iHYCdTAOqF*

## Abstract

Time-series forecasting is an important task in many domains, including finance, weather prediction, and energy consumption forecasting, and deep learning methods have emerged as the best-performing time-series forecasting methods over the last few years. However, most proposed time-series forecasting models are deterministic and are prone to errors when deployed in production, potentially causing significant losses and penalties when making predictions with low confidence. In this paper, we propose the Time-Energy Model (TEM), a framework that introduces so-called *selective time-series forecasting* using energy-based models (EBMs). Selective forecasting estimates model confidence and allows the end-user to selectively reject forecasts while maintaining a desired target coverage. TEM is model-agnostic and can be used to improve forecasting accuracy of any encoder-decoder deterministic time-series forecasting model. TEM is trained using a combination of supervised and self-supervised learning, leveraging excellent single-point prediction accuracy while maintaining the ability to reject forecasts based on model confidence. Experimental results indicate that TEM generalizes well across 5 state-of-the-art deterministic time-series forecasting models and 5 benchmark time-series forecasting datasets. Using selective forecasting, TEM reduces prediction error by up to 49.1% over 5 state-of-the-art deterministic models. Furthermore, TEM has up to 87.0% lower error than selected baseline EBM models, and achieves significantly better performance than state-of-the-art selective deep learning models. Code for the proposed TEM framework is available at https://github.com/JonasBrusokas/Time-Energy-Model.

## 1 Introduction

Time-series forecasting plays a pivotal role in various domains, enabling informed decisions such as smart building control, adjusting the operation of heating systems, and buying and selling financial assets (Jin et al., 2021; Affonso et al., 2021). Recent advancements in time-series forecasting using deep learning have significantly improved prediction accuracy and efficiency while addressing key limitations of previous models,

such as high computational costs and inability to capture global time-series patterns (Zhou et al., 2021; Wu et al., 2021; Zhou et al., 2022; Wu et al., 2023; Nie et al., 2023). Most current time-series forecasting models are deterministic, producing a single prediction for an observed target process using historical data as input. However, single predictions are often insufficient for real-world applications, as they do not estimate model confidence that would enable decision makers to avoid using inaccurate predictions (Wen et al., 2017; Gneiting, 2011).

*Selective prediction, also known as prediction with a reject option*, addresses these limitations by allowing models to abstain from making predictions when model confidence is low. Enabling predictive models to reject potentially inaccurate predictions provides significant utility in domains where prediction errors carry significant costs (Lathe & Saeys, 2024; Hasan et al., 2023). Recent applications of selective prediction include healthcare diagnostics, autonomous driving systems, and financial markets (Zhang et al., 2023; Mohri et al., 2024; Cao et al., 2024). Despite their benefits, selective prediction has not been explored in the context of time-series forecasting, and although there are similar neural network architectures enabling selective prediction for time-series *classification*, there are no known selective time-series forecasting methods (Nam et al., 2022; Zhang et al., 2023).

The only applicable selective prediction framework for time-series forecasting is SelectiveNet, which was developed for classification and regression tasks (Geifman & El-Yaniv, 2019). SelectiveNet enables selective prediction based on user-defined target coverage, which describes the minimum number of predictions the model should perform. This framework provides a specialized loss function based on the interior point optimization method and defines neural network architectures for both classification and regression. However, due to SelectiveNet's loss function, the models suffer from degraded prediction accuracy compared to deterministic models. SelectiveNet also requires training a separate model for each user-defined target coverage and does not allow for rejecting predictions based on other criteria, such as estimated prediction error.

Energy-based models (EBMs) have been extensively studied and have recently seen a resurgence in the machine learning community. EBMs are unnormalized probabilistic models that provide a scalar measure called *energy* estimating the compatibility between a given input and output. EBMs provide an unnormalized density over all configurations of input and output with lower energy being assigned to more likely configurations. EBMs make no prior assumptions about the output and are capable of capturing highly complex output distributions (Gustafsson et al., 2022). Recently, EBMs parameterized by deep neural networks have been successfully applied to various machine learning tasks (Gustafsson et al., 2020; Hendriks et al., 2021; Gustafsson et al., 2021; Castillo-Navarro et al., 2022; Tu et al., 2020b;a; Li et al., 2021; Zhu et al., 2024; Singh et al., 2024; Li et al., 2023). Time-series EBMs could therefore be used for *selective time-series forecasting*, where the energy for a given input and output could be used to estimate model confidence and selectively reject inaccurate predictions without having to retrain the entire model.

However, applying EBMs for time-series forecasting has several open challenges: (i) To be applied for time-series forecasting, an EBM architecture and training method must be capable of capturing and learning sequential dependencies and provide accurate predictions. Many recent time-series forecasting papers propose more accurate and efficient architectures for time-series forecasting (Zhou et al., 2021; Wu et al., 2021; Zhou et al., 2022; Wu et al., 2023; Nie et al., 2023). However, there are no currently known EBM architectures for time-series. Generative methods, such as EBMs, tend to quantitatively underperform against tailor-made discriminative methods in downstream tasks, such as classification, regression, and forecasting (Grathwohl, 2021; Zheng et al., 2023). Accurate predictions are key to providing utility in decision making, thus an EBM for time-series forecasting must have comparable performance to deterministic forecasting models. (ii) Inference on time-series using an EBM must be scalable for arbitrary time-series forecasting horizons. Many EBM inference methods rely on deterministic sampling techniques, generating outputs autoregressively by exploring a sufficient subset of the output space to produce an accurate prediction. However, these techniques are generally computationally expensive for time-series as both the input and output spaces scale exponentially with series length, making the solution space too large to traverse in a reasonable time (Gustafsson et al., 2020; Tu et al., 2020b).

This paper addresses these challenges by proposing an energy-based model framework for time-series forecasting called the *Time-Energy Model* (TEM). This paper makes the following contributions: (1) Proposes

the *Time-Energy Model* (TEM), an energy-based model framework for time-series forecasting parameterized by deep neural networks addressing challenge (i). (2) Proposes joint parameterization and training techniques of state-of-the-art plug-in deterministic models and energy-based models for time-series forecasting, combining high single prediction accuracy with estimating prediction error addressing challenge (i). (3) Proposes a scalable selective prediction procedure: *selective forecasting* with 2 inference methods that use the energy-based model to estimate prediction error and enable rejecting predictions above a selected error bound, addressing challenges (i, ii). (4) Provides an experimental quantitative and qualitative evaluation of TEM on 5 benchmark time-series datasets, 1 baseline energy-based model for regression, 1 state-of-the-art selective prediction deep learning model, and 5 state-of-the-art deterministic forecasting models. Using selective forecasting, TEM reduces prediction error by up to 49.1% over 5 state-of-the-art deterministic models. The evaluation shows that TEM has up to 87% lower prediction error than applicable EBM models for time-series and over 4244.3% lower error than baseline SelectiveNet selective prediction models. The experiments demonstrate that TEM generalizes to improve the accuracy of all 5 selected plug-in encoder-decoder deterministic time-series forecasting models over 5 datasets.

The remainder of this paper is structured as follows: Section 2 provides the problem definition and background on energy-based models and selective forecasting. Section 3 presents the proposed TEM framework, including its architecture, training procedure, and selective forecasting methods. Section 4 describes the experimental setup, including baseline forecasting models, benchmark datasets, and evaluation metrics. Section 5 presents and analyzes the experimental results. Finally, Section 6 concludes the paper and discusses future work directions.

## 2 Problem Definition

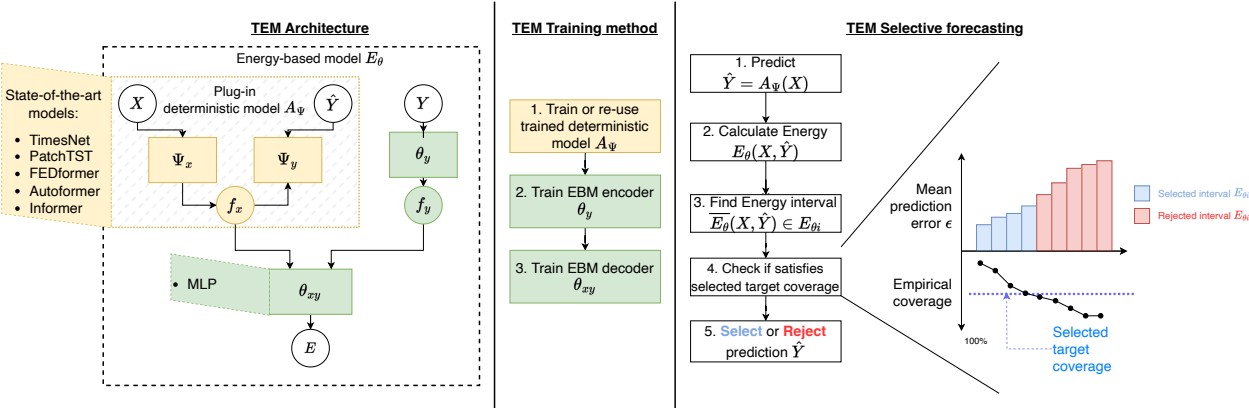

Figure 1: Overview of the TEM framework. Starting from the left: TEM architecture, TEM training method, and TEM Selective Forecasting. Colors indicate which TEM components are trained at which step.

This section formally describes the preliminaries and provides a problem definition for the paper.

*Deterministic time series forecasting.* $X$ is a regular multivariate time series containing observed data needed for prediction with sequence length $m$. $X$ is composed of vectors $z_t, z_t \in \mathbb{R}^d$ representing $d$ observed features at time step $t$. $\mathcal{X}$, where $\mathcal{X} = \mathbb{R}^{m \times d}$, $X \in \mathcal{X}$, is the space containing all possible input time-series.

$$X = \begin{bmatrix} z_{t-m+1} & \dots & z_{t-1} & z_t \end{bmatrix} \quad (1) \qquad Y = \begin{bmatrix} y_{t+1} & \dots & y_{t+h-1} & y_{t+h} \end{bmatrix} \quad (2)$$

$Y$ is a regular time series containing values for a single observed feature ahead of time step $t$. Prediction horizon $h$ defines how many time steps ahead will be predicted. $Y$ is composed of real numbers $y_t, y_t \in \mathbb{R}$ representing the observed feature. In this paper $\hat{Y}, \hat{Y} \in \mathcal{Y}$ is a single best-guess prediction. $\mathcal{Y}$, where $\mathcal{Y} = \mathbb{R}^h$, $Y \in \mathcal{Y}$, is the space containing all possible output time-series, such that a mapping $\mathcal{X} \rightarrow \mathcal{Y}$ exists.

*Deterministic time-series forecasting models.* A deterministic time-series forecasting model $A_\Psi$ is a predictive model with parameters $\Psi$ that takes $X$ as an argument and produces a single $\hat{Y}$ prediction as output. The prediction error $\epsilon$ is defined as the absolute difference between observed output $Y$ and prediction $\hat{Y}$.

*Energy-based models.* An energy-based model $E_\theta$ is a predictive model with parameters $\theta$ that takes $X$ and $Y$ as arguments and produces energy $E$ as output. $E \in \mathbb{R}$ is a measure of compatibility between given $X$ and $Y$, where lower energy means higher compatibility (LeCun et al., 2006).

$$A_\Psi : \mathcal{X} \to \mathcal{Y}, \qquad E_\theta : \mathcal{X} \times \mathcal{Y} \to \mathbb{R} \tag{3}$$

$E_\theta$ can be defined as an unnormalized probabilistic model, defining a conditional distribution $\rho_\theta(Y|X)$ for possible output $Y$, given $X$.

$$\rho_\theta(Y|X) = \frac{\exp(-E_\theta(X,Y))}{\int_Y \exp(-E_\theta(X,Y)) \ = \ Z(\theta)} \qquad (4) \qquad\qquad \hat{Y} = \arg\min_Y E_\theta(X,Y) \tag{5}$$

The normalization constant $Z(\theta)$ is intractable in a general case. However, calculating $Z(\theta)$ is not strictly necessary for energy-based model training or inference. Unlike deterministic models, inference using an EBM is done by finding predicted output $\hat{Y}$ that minimizes energy w.r.t. $\hat{Y} \in \mathbb{Y}$, given $X$.

*Selective forecasting.* Selective prediction for time-series (selective forecasting) can be defined as a pair of a deterministic forecasting model $A_\Psi$ and selection function $g$. The deterministic forecasting model $A_\Psi$ provides predictions $\hat{Y}$ for given input $X$ (as defined in Equation 3), where the selection function $g$ is a decision function for selecting or rejecting the prediction $\hat{Y}$.

$$(A_\Psi, g)(X) \triangleq \begin{cases} A_\Psi(X) = \hat{Y} & \text{if } g = 1, \\ \text{None}, & \text{if } g = 0, \end{cases} \tag{6}$$

Selective prediction performance can be quantified using *selective coverage* and *selective risk*. Selective coverage $\phi(g)$ quantifies the proportion of selected predictions using selection function $g$. Coverage can also be viewed as the probability of a prediction being selected using function $g$.

Selective risk $R(A_\Psi, g, l)$ defines prediction error for predictive model $A_\Psi$ for selected predictions using selection function $g$. Prediction error is calculated with distance metric $l$, such as Mean Squared Error (MSE).

$$\phi(g) \triangleq \mathbb{E}[g(x)] \equiv P(g = 1) \qquad (7) \qquad\qquad R(A_\Psi, g, l) = \frac{\mathbb{E}\big[l\big((A_\Psi, g)(X), Y\big) \cdot g(X)\big]}{\phi(g)} \tag{8}$$

*Problem definition.* Given time series $X, Y$, find an energy-based model $E_\theta$ with parameterization $\theta$ jointly trained with a deterministic model $A_\Psi$ and selection function $g$, such that selective risk $R(A_\Psi, g, l)$ is minimized while controlling selective coverage $\phi(g)$.

## 3 TIME-ENERGY MODEL (TEM)

In this paper, we propose TEM, a deep-learning framework for time-series forecasting using energy-based models that enables selective prediction based on user-defined target coverage.

### 3.1 TEM overview

TEM is a novel energy-based time-series forecasting framework that combines the accuracy and low latency of deterministic forecasting models with selective forecasting capabilities using an energy-based model. As shown in Figure 1, TEM consists of three key components: 1) TEM architecture 2) TEM joint training method 3) TEM selective forecasting.

The TEM architecture consists of an encoder-decoder-based plug-in deterministic forecasting model $A_\Psi$ that provides accurate low-latency forecasts $\hat{Y}$ and an energy-based model $E_\theta$ that reuses the parameters of $A_\Psi$ for estimating energy $E(X, Y)$. Both $A_\Psi$ and $E_\theta$ learn to capture sequential dependencies in time-series $X, Y$. The TEM framework allows "plugging in" any deterministic encoder-decoder forecasting model $A_\Psi$ (Section 3.2). TEM provides a training method that utilizes both supervised and self-supervised learning to jointly train $A_\Psi$ and $E_\theta$ (Section 3.3). TEM introduces selective forecasting which uses the energy-based model $E_\theta$ to achieve user-defined target coverage while minimizing selective risk (Section 3.4).

## 3.2 TEM architecture

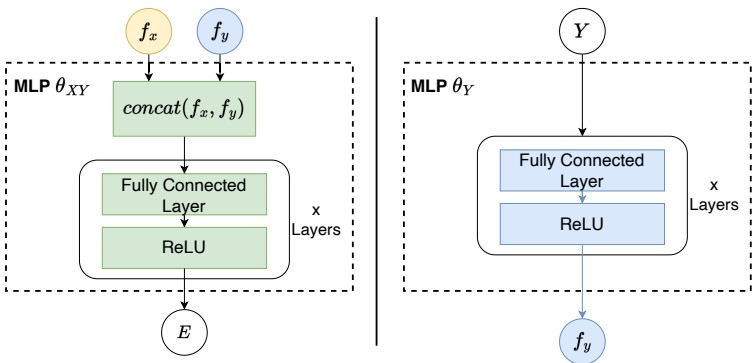

Figure 2: Proposed architectures for TEM EBM encoder $\theta y$ and decoder $\theta xy$. Starting from the left: the MLP-based architecture for TEM decoder $\theta xy$, the MLP-based architecture for TEM encoder $\theta y$.

As shown in Figure 1, TEM consists of two interoperating deep neural network time-series models: the encoder-decoder plug-in deterministic forecasting model $A_\Psi$ and the energy-based model $E_\theta$.

*Plug-in deterministic model $A_\Psi$.* A deterministic encoder-decoder-based plug-in forecasting model $A_\Psi$ is parameterized by $\Psi$ and is composed of an input encoder $\Psi_x$ and an input decoder $\Psi_y$.

$$A_\Psi : \Psi_y(\Psi_x(X)) \mapsto \hat{Y} \quad (9) \qquad\qquad E_\theta : \theta xy(\Psi_x(X), \theta y(Y)) \mapsto E \quad (10)$$

As shown in Equation 9, the input encoder $\Psi_x$ is trained to produce a hidden input representation $f_x$ from an input $X$. $f_x$ is then used by the input decoder $\Psi_y$ to produce an accurate forecast $\hat{Y}$. Most recent state-of-the-art transformer-based deterministic forecasting models use encoder-decoder architectures (Wen et al., 2023; Zhou et al., 2021; Wu et al., 2021; Zhou et al., 2022; Nie et al., 2023; Wu et al., 2023).

*Energy-based model $E_\theta$.* The energy-based model $E_\theta$ is parameterized by $\theta$. $E_\theta$ re-uses the encoder $\Psi_x$ from the deterministic forecasting model $A_\Psi$ to calculate the hidden input representation $f_x$. $E_\theta$ consists of an output encoder $\theta y$ and an output decoder $\theta xy$, using the joint representation of $f_x$ and $f_y$ to calculate energy. As shown in Equation 10, the output encoder $\theta y$ produces a hidden output representation $f_y$ from an arbitrary given output $Y$. The output decoder $\theta xy$ then uses representations $f_x$ and $f_y$ to produce energy $E = E_\theta(X, Y)$.

Like the deterministic model $A_\Psi$, the EBM encoder and decoder $\theta y, \theta xy$ are parameterized using deep neural networks. As shown in Figure 2, we propose using a multi-layer perceptron (MLP) architecture for $\theta y$ to calculate the hidden output representation $f_y$. MLPs provide very low computational latency, enabling fast energy calculation for different $Y$ values and faster inference. For $\theta xy$, we propose an architecture similar to $\theta y$ that concatenates the two representations $f_x, f_y$ and uses an MLP decoder to produce energy $E$.

## 3.3 TEM training

TEM uses a joint training method that combines supervised and self-supervised learning techniques to train both the deterministic forecasting model $A_\Psi$ and the EBM $E_\theta$.

*Training method overview.* We propose using traditional supervised learning to train the state-of-the-art forecasting model parameters $\Psi$ and then using self-supervised learning to train the $E_\theta$ parameters. As shown in Figure 1, three components are trained in the following sequence:

1. Training $A_\Psi$. The deterministic encoder-decoder forecasting model is trained with supervised learning using the loss function and hyperparameters as described in known literature. After training, $A_\Psi$ parameters $\Psi_y, \Psi_x$ are frozen. Alternatively, if $A_\Psi$ is trained apriori, we can directly reuse the model parameters $\Psi$.

2. Training $E_\theta$ parameters $\theta y$ and $\theta xy$. The EBM encoder $\theta y$ and decoder $\theta xy$ are trained using Contrastive Divergence self-supervised learning (Hinton, 2002). Although the EBM $E_\theta$ uses the encoder $\Psi_x$ from the deterministic model $A_\Psi$ (as shown in Equation 10), the parameters of the deterministic model $\Psi$ remain frozen during this step.

This training method preserves the state-of-the-art forecasting accuracy of deterministic forecasting models $A_\Psi$ while enabling energy calculation $E_\theta(X, Y)$ for forecasts using the EBM $E_\theta$.

*EBM training with Contrastive Divergence.* Contrastive Divergence (CD) is a parameter estimation method for learning EBMs (Hinton, 2002; Song & Kingma, 2021).

CD learns the EBM parameters by contrasting a "positive" output sample $Y^{(0)}$ from the training set for given $X$ against a single "negative" sample $Y^{(1)}$.

$$Y^{(1)} = Y^{(1)} - \eta \nabla_{Y^{(1)}} E_\theta(X, Y^{(1)}) + \omega \tag{11}$$

As shown in Equation 11, the negative sample $Y^{(1)}$ is obtained by refining a randomly generated point using Langevin dynamics (initialized from $\mathcal{N}(0, \sigma^2 I)$) with step size $\eta$ and step count $N_{CD}$.

$$\mathcal{L}_{CD} = (E^+ - E^-) + \alpha_{CD}((E^+)^2 + (E^-)^2) \tag{12}$$

As shown in Equation 12, the loss $\mathcal{L}_{CD}$ is calculated as the difference between positive and negative sample energies $E_\theta(X, Y^{(0)}) - E_\theta(X, Y^{(1)})$, with a regularization term multiplied by coefficient $\alpha_{CD}$. A detailed description of the CD training method is provided in Appendix Algorithm 1.

## 3.4 Selective forecasting with TEM

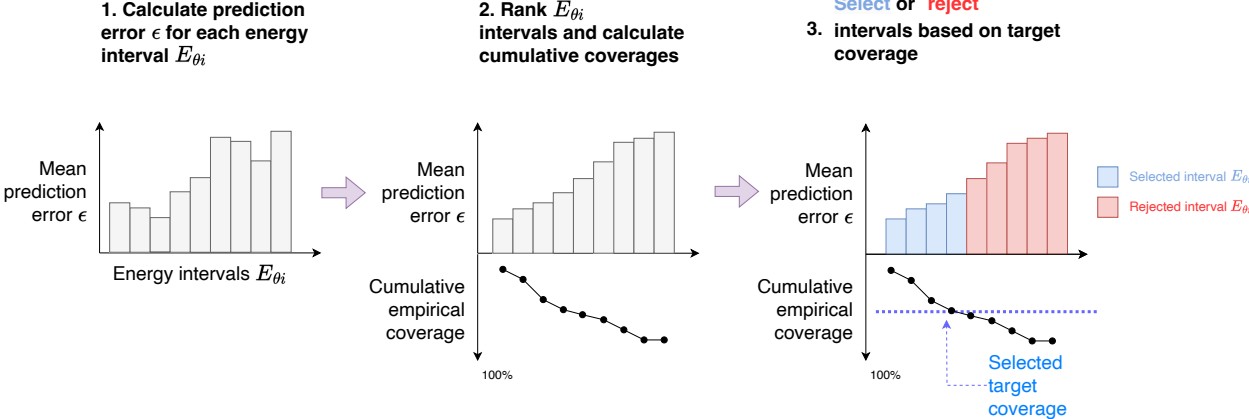

Figure 3: Overview of selective forecasting with TEM: (1) Energy values are calculated for training set predictions and partitioned into intervals. (2) Mean forecast error and empirical coverage are calculated for each interval. (3) Intervals are ranked by error and selected to achieve desired target coverage.

TEM proposes a novel method to perform selective forecasting by using energy values to maintain user-defined target coverage while minimizing selective risk. It uses the EBM $E_\theta$ to evaluate forecasts $\hat{Y}$ made by the deterministic model $A_\Psi$ and selectively reject forecasts.

In traditional (normalized) probabilistic forecasting, achieving user-defined target coverage can be done using parameters of the estimated output probability distribution, quantiles, or generated samples (Salinas et al., 2019; Wen et al., 2017). However, an EBM $E_\theta$ is an unnormalized probabilistic model and the computed energy $E = E_\theta(X, Y)$ can only be interpreted as a relative compatibility score for a given input and forecast pair $X, Y$. Thus, energy scores can only be directly compared for the same input $X$, with exact score values varying between different $X$ values. To circumvent this, other literature proposes deterministically sampling energy values around $Y$ to evaluate forecasts (Gustafsson et al., 2020). For time-series, the output space grows exponentially with forecast horizon, making it infeasible to deterministically sample the output space in a reasonable time.

TEM proposes to use both the energy of the single forecast $E_\theta(X, \hat{Y})$ and the energy values for $\hat{Y} + \delta \simeq \hat{Y}$ around the forecast $\hat{Y}$ to rank forecasts and achieve target coverage. If the initial forecast $\hat{Y}$ is accurate, the energy values around the forecast $E_\theta(X, \hat{Y} + \delta)$ should be low. Alternatively, if the initial forecast $\hat{Y}$ is not accurate or the output distribution has high variance or is highly multimodal, the energy around the forecast should be higher. We propose two inference methods for achieving target coverage with TEM: 1) Aggregated energy inference – which ranks forecasts by adding noise to the forecasts $\hat{Y}$ to sample energy and 2) Energy optimization inference – which ranks forecasts by minimizing the energy for forecast $\hat{Y}$.

*Aggregated energy inference.* The Aggregated energy inference method directly samples the energy values around the prediction $\hat{Y}$. These energy values are used to calculate *aggregated energy* which is then related to prediction error $\epsilon$. Aggregated energy $\overline{E_\theta}(X, \hat{Y})$ can be defined as:

$$\overline{E_\theta}(X, \hat{Y}) = \frac{\sum_{i=1}^{n} E_\theta(X, \hat{Y} + \delta_i)}{n} - E_\theta(X, \hat{Y}), \tag{13}$$

where $\overline{E_\theta}(X, \hat{Y})$ is the the mean of energy values $E_\theta(X, \hat{Y} + \delta_i)$ calculated on and around $\hat{Y}$ by adding noise $\delta_i$. Samples $\delta_i$ are drawn from a noise distribution. In this paper, we will use the multivariate normal distribution $\mathcal{N}(0, \sigma^2 I)$, where $I$ is an identity matrix and covariance coefficient $\sigma^2$ is selected according to the model and data.

*Energy optimization inference.* We also propose an alternative inference method called Energy optimization inference. This inference method is based on traditional EBM inference methods (as shown in Equation 5) and directly minimizes the energy $E_\theta(X, \hat{Y})$ on the prediction $\hat{Y}$ and then relates the energy to prediction error $\epsilon$. Energy optimization inference can be defined as:

$$\hat{E}(X, \hat{Y}) = E_\theta(X, Y), \text{where } Y = \underset{Y}{\arg\min}\, E_\theta(X, Y) \tag{14}$$

where $\hat{E}(X, \hat{Y})$ is the minimized energy $E_\theta(X, Y)$ w.r.t. $Y$. The initial value for $Y$ is the single prediction $\hat{Y}$ made by the deterministic model $A_\Psi$. We use gradient descent to minimize $E_\theta(X, Y)$ with step count $T$ and step size $\eta$. Unlike the Aggregated energy inference method, the Energy optimization method relates a single energy value $\hat{E}(X, \hat{Y})$ to prediction error $\epsilon$. Notably, TEM weights $\theta$ are not updated while minimizing $E_\theta(X, Y)$.

*Using energy for selective forecasting.* As shown in Figure 3, we propose to calibrate TEM using energy to estimate model confidence and select forecasts with the lowest error $\epsilon$ , while still achieving the desired user-defined target coverage $\phi(g)$. To achieve this, we partition the energy range into a finite number $R$ of disjoint energy intervals $E_{\theta i}$ and calculate the mean forecast error $\epsilon$ and empirical coverage $\phi(g)$ of all training $X, Y$ samples for which $E_\theta(X, \hat{Y}) \in E_{\theta i}$, as shown in step 1 in Figure 3. As shown in step 2 in Figure 3, we rank all energy intervals in ascending order of forecast error $\epsilon$ and calculate the cumulative empirical coverage for each interval starting from the interval with the lowest forecast error. As shown in step 3 in Figure 3, we then calculate which energy intervals should be selected or rejected for a desired target coverage $\phi(g)$.

After TEM calibration, the model forecasts can be selected or rejected by calculating the energy value $E_\theta(X, \hat{Y})$ of the prediction $\hat{Y}$ and checking which energy interval the energy belongs to $E_\theta(X, \hat{Y}) \in E_{\theta i}$. Notably, the target coverage $\phi(g)$ can be dynamically selected based on the utility of the prediction and the requirements of the application without the need to retrain or recalibrate the model $E_\theta$. Furthermore, this approach enables forecasts to be selected or rejected based on other criteria, such as prediction error $\epsilon$.

## 4 Experimental Setup

### 4.1 Baseline models

*Baseline Energy-based models.* In this paper, we use EB-NARX (Hendriks et al., 2021) as a baseline energy-based model to evaluate the performance of the TEM framework. EB-NARX is an energy-based model parameterized by deep neural networks that was initially developed for time-series regression. While EB-NARX uses a combination of solution space sampling and energy minimization w.r.t. $Y$ to perform inference, this method is not scalable for multi-step time-series forecasting as the output space grows exponentially with forecast horizon. We perform multi-step forecasting with EB-NARX by making predictions one time-step at a time (from $y_{t+1}$ to $y_{t+h}$), propagating each single best-guess prediction until reaching the desired forecast horizon.

*State-of-the-art deterministic forecasting models.* In this paper, we use 5 state-of-the-art transformer-based deterministic time-series forecasting models to evaluate the performance of the TEM framework.

Informer (Zhou et al., 2021) was one of the first Transformer models for deterministic time-series forecasting. Informer uses direct multi-step inference avoiding error accumulation in the autoregressive forecasting setting. It also was one of the first such models to utilize learnable positional encodings for input sequence and max-pooling to down-sample intermediate hidden representations (Wen et al., 2023).

Autoformer (Wu et al., 2021) built on Informer by introducing seasonal trend decomposition and a novel autocorrelation block instead of a traditional attention module reducing inference complexity while providing higher prediction accuracy (Wen et al., 2023).

FEDformer (Zhou et al., 2022) further built on Informer and Autoformer by introducing Fourier and Wavelet transformations in addition to seasonal decomposition. It achieves a higher prediction accuracy with significantly lower inference and memory complexity (Wen et al., 2023).

TimesNet (Wu et al., 2023) proposes a novel approach that treats time series forecasting as an image-to-image translation problem. It introduces a learnable time-frequency transformation to capture both temporal and frequency patterns in time series data. TimesNet also utilizes a series of inception blocks with different kernel sizes to capture multi-scale temporal dependencies, allowing it to adapt to various seasonal patterns and trends in time series data.

PatchTST (Nie et al., 2023) is a patch-based time series transformer that addresses the limitations of previous transformer models in capturing long-range dependencies. PatchTST divides the input time series into non-overlapping patches and applies self-attention mechanisms to efficiently process longer input sequences and capture both local and global temporal patterns. These techniques result in significantly improved forecasting performance for long-term predictions.

*SelectiveNet-like time-series forecasting models.* SelectiveNet (Geifman & El-Yaniv, 2019) is a deep learning framework proposed for enabling selective prediction with neural networks. It uses a specialized selective loss function and a neural network architecture that produces three outputs: selection, prediction, and auxiliary prediction. These outputs are used to train the model and to perform selective prediction - reject a proportion of predictions to achieve the desired target coverage $\phi(g)$. Each SelectiveNet model is trained for a specific target coverage $\phi(g)$, that is set before training the model. To the best of our knowledge there is no SelectiveNet implementation for selective time-series forecasting. As such, we adapt the framework to use selective loss based on MSE (which is used as a loss function by state-of-the-art deterministic forecasting models described in 4.1) SelectiveNet was selected as a baseline as it is the only end-to-end deep learning

framework enabling coverage-based selective prediction that *could* be applicable for selective time-series forecasting.

## 4.2 Datasets

To evaluate TEM performance, we use five open benchmark time-series datasets. The Electricity Transformer Temperature datasets: ETTh1, ETTh2 (Zhou et al., 2021) contain 2 years of hourly temperature measurements from two electricity transformers in separate Chinese counties, each with 7 sensor features. The Exchange Rate dataset contains the daily exchange rates between 8 different currencies against USD from 1990 to 2016, with XRP/USD as the target variable for forecasting. The Weather dataset contains 4 years of daily weather measurements from 21 monitoring stations across Canada, with the target variable being the temperature readings from a specific station. The National Illness dataset contains weekly influenza-like illness ratios reported by the US Centers for Disease Control (CDC), containing data from 2002 to 2021 across multiple US regions. These datasets were selected as they are commonly used to benchmark time-series forecasting models and we re-use the data preprocessing and splitting procedures, as found in recent state-of-the-art deterministic forecasting model literature (Zhou et al., 2021; Wu et al., 2021; Zhou et al., 2022; Wu et al., 2023; Nie et al., 2023). Additional statistics for the datasets are provided in Appendix 6.

## 4.3 Metrics

We use the Mean Square Error (MSE) metric to evaluate prediction error for all forecasting models. To evaluate selective forecasting performance for both TEM and SelectiveNet, we use selective coverage $\phi(g)$ and selective risk $R(A_\Psi, g, l)$ with MSE as the distance metric (as defined in Equations 7, 8).

## 4.4 Implementation details

For the deterministic forecasting models Informer, Autoformer, FEDformer, PatchTST, and TimesNet $A_\Psi$ we re-use known hyperparameters from their respective experiments (Zhou et al., 2021; Wu et al., 2021; Zhou et al., 2022; Wu et al., 2023; Nie et al., 2023). For the MLP-based encoder and decoder $\theta_y, \theta_{xy}$ parameterizations, we use 4 layers for each with 128 hidden units in each of the fully connected layers.

For TEM selective forecasting using Aggregated energy inference, as described in Section 3.4, we select the covariance coefficient $\sigma^2$ for the multivariate normal distribution $\mathcal{N}(0, \sigma^2 I)$ from which we will draw noise samples $\delta_i$. We select one $\sigma^2 \in \{0.0, 0.02, 0.05, 0.1, 0.2, 0.3, 0.5\}$ for each trained TEM model. For each prediction $\hat{Y}$, we draw 32 samples $\delta_i$ to generate aggregated energy $\overline{E_\theta}(X, \hat{Y})$. For TEM selective forecasting using Energy optimization inference, we perform gradient descent using the Adam optimizer using step sizes $\eta \in \{0.1, 0.01, 0.001\}$ and step counts $T \in \{5, 10, 25\}$. We provide additional implementation details in Appendix Section A.6. Changing TEM selective forecasting parameters does not require changing or retraining any components of TEM.

# 5 Results

## 5.1 Quantitative TEM Selective forecasting performance

We evaluate TEM models in terms of selective risk and coverage performance with target coverage $\phi(g)$ against plug-in deterministic models $A_\Psi$. We quantitatively evaluate TEM using all combinations of: 5 datasets (Section 4.2), 5 plug-in deterministic forecasting models $A_\Psi$ (Section 4.1), and 2 TEM inference methods (Section 3.4).

For each configuration, we conducted 3 experiments with different random number seeds to reduce the likelihood of non-representative results. Selected target coverages $\phi(g) \in \{10\%, 30\%, 50\%, 70\%, 90\%\}$ were chosen to include those used in the SelectiveNet paper (Geifman & El-Yaniv, 2019) as well as 10% and 30%.

*Overall TEM performance.* As shown in Table 1, TEM reduces prediction error across all configurations of five deterministic models $A_\Psi$ and five benchmark datasets. Tables highlight the best performing models for each target coverage, by choosing the model with the lowest prediction error for each target coverage, if the

Table 1: TEM performance comparison across different models and datasets. Results show selective risk and empirical coverage (in parentheses) for target coverages $\phi(g) \in 10\%, 30\%, 50\%, 70\%, 90\%$. Best performing models for specific target coverages are marked **bold**.

| Base Model | Coverage | Method | Dataset | | | | |
|---|---|---|---|---|---|---|---|
| | | | ETTh1 | ETTh2 | Weather | Exchange Rate | National Illness |
| | Original | - | 0.0876 | 0.1577 | 0.0079 | 0.0899 | 1.1758 |
| | 10% | TEM | **0.0906 (16.80)** | **0.1591 (28.39)** | **0.0076 (57.13)** | **0.0712 (17.60)** | **0.5634 (2.78)** |
| | | SelectiveNet | 0.7662 (55.92) | 0.7103 (40.13) | 0.2903 (45.41) | 1.1609 (60.56) | 3.3422 (51.91) |
| | 30% | TEM | **0.0889 (51.79)** | **0.1591 (28.39)** | **0.0076 (57.13)** | **0.0708 (26.73)** | **1.0141 (20.14)** |
| | | SelectiveNet | 0.6855 (55.45) | 0.9926 (54.14) | 0.3479 (54.13) | 1.4727 (52.55) | 3.6843 (56.43) |
| Autoformer | 50% | TEM | **0.0889 (51.79)** | **0.1568 (59.22)** | **0.0076 (57.13)** | **0.0902 (61.93)** | **1.0128 (42.13)** |
| | | SelectiveNet | 0.6905 (54.08) | 1.0288 (55.82) | 0.3276 (51.06) | 1.7034 (58.36) | 4.1128 (57.38) |
| | 70% | TEM | **0.0871 (78.64)** | **0.1568 (59.22)** | **0.0075 (73.91)** | **0.0897 (69.35)** | **1.0464 (65.74)** |
| | | SelectiveNet | 0.9668 (79.58) | 1.5178 (81.07) | 0.6144 (72.30) | 1.3532 (76.79) | 5.1194 (75.45) |
| | 90% | TEM | **0.0881 (87.45)** | **0.1568 (76.78)** | **0.0075 (73.91)** | **0.0890 (87.96)** | **1.0974 (83.10)** |
| | | SelectiveNet | 1.5981 (91.95) | 1.5979 (84.55) | 0.6439 (96.85) | 3.1883 (93.54) | 5.6797 (84.31) |
| | Original | - | 0.0772 | 0.1184 | 0.011 | 0.0653 | 1.0503 |
| | 10% | TEM | **0.0782 (45.86)** | **0.0958 (17.75)** | **0.0104 (61.27)** | **0.0354 (3.19)** | **0.6754 (4.40)** |
| | | SelectiveNet | 0.5129 (25.05) | 0.6745 (19.00) | 0.3382 (49.98) | 0.9054 (48.13) | 4.1730 (97.74) |
| | 30% | TEM | **0.0782 (45.86)** | **0.1192 (75.68)** | **0.0104 (61.27)** | **0.0616 (34.74)** | **0.8927 (13.66)** |
| | | SelectiveNet | 1.1763 (63.16) | 3.0803 (86.68) | 0.3297 (49.42) | 0.3431 (19.02) | 3.4399 (80.57) |
| FEDformer | 50% | TEM | **0.0770 (94.77)** | **0.1192 (75.68)** | **0.0104 (61.27)** | **0.0616 (34.74)** | **0.8470 (26.39)** |
| | | SelectiveNet | 0.5933 (30.61) | 0.6300 (18.90) | 0.4021 (59.81) | 1.3906 (76.61) | 1.9532 (44.76) |
| | 70% | TEM | **0.0770 (94.77)** | **0.1192 (75.68)** | **0.0105 (67.49)** | **0.0634 (72.38)** | **0.8748 (43.75)** |
| | | SelectiveNet | 1.1240 (70.69) | 3.3039 (91.68) | 0.5702 (85.10) | 1.4646 (75.63) | 2.5126 (61.66) |
| | 90% | TEM | **0.0770 (94.77)** | **0.1190 (85.77)** | **0.0105 (67.49)** | **0.0655 (91.33)** | **0.9229 (62.73)** |
| | | SelectiveNet | 1.8596 (97.29) | 3.7828 (96.10) | 0.6625 (98.34) | 1.6270 (93.52) | 3.8254 (86.50) |
| | Original | - | 0.6461 | 1.1877 | 0.3313 | 0.73 | 4.6609 |
| | 10% | TEM | **0.6008 (37.08)** | 1.2057 (13.05) | **0.0046 (54.75)** | **0.1828 (7.81)** | 4.0733 (9.03) |
| | | SelectiveNet | 0.7258 (43.89) | **0.4411 (13.14)** | 0.2557 (38.93) | 0.7313 (38.84) | **2.4625 (51.77)** |
| | 30% | TEM | **0.6008 (37.08)** | 1.1754 (27.66) | **0.0046 (54.75)** | **0.2000 (8.11)** | 4.4375 (47.92) |
| | | SelectiveNet | **0.5989 (37.10)** | **0.4349 (13.82)** | 0.1355 (21.43) | 1.6538 (91.89) | **1.8485 (45.88)** |
| Informer | 50% | TEM | **0.6460 (89.32)** | 1.1845 (49.45) | **0.0046 (54.75)** | **0.1921 (10.47)** | 4.3605 (66.20) |
| | | SelectiveNet | 0.8827 (61.93) | **0.4902 (14.88)** | 0.3819 (56.73) | 0.0723 (3.91) | **2.1334 (39.48)** |
| | 70% | TEM | **0.6460 (89.32)** | 1.1814 (70.75) | **0.0046 (59.67)** | 0.2818 (12.98) | 4.5612 (80.32) |
| | | SelectiveNet | 1.5353 (93.71) | 2.3146 (81.68) | 0.6279 (95.77) | 0.0774 (3.85) | **3.3572 (82.48)** |
| | 90% | TEM | **0.6438 (93.87)** | 1.1850 (85.72) | **0.0046 (59.67)** | **0.4741 (34.02)** | 4.5576 (90.74) |
| | | SelectiveNet | 1.5991 (93.58) | 2.7387 (90.36) | 0.6709 (99.76) | 2.5347 (97.99) | 4.7834 (96.82) |
| | Original | - | 0.0416 | 0.1079 | 0.0011 | 0.0617 | 0.7324 |
| | 10% | TEM | **0.0340 (15.32)** | **0.0735 (11.98)** | **0.0008 (21.02)** | **0.0488 (6.33)** | **0.4686 (6.25)** |
| | | SelectiveNet | 0.7245 (46.66) | 1.1741 (43.79) | 0.2904 (44.02) | 1.3008 (43.17) | 3.0678 (41.82) |
| | 30% | TEM | **0.0413 (30.05)** | **0.1077 (62.31)** | **0.0008 (21.02)** | **0.0526 (7.60)** | **0.4209 (12.96)** |
| | | SelectiveNet | 0.7245 (46.66) | 1.1741 (43.79) | 0.2904 (44.02) | 1.3008 (43.17) | 3.0678 (41.82) |
| PatchTST | 50% | TEM | **0.0418 (55.94)** | **0.1077 (62.31)** | **0.0008 (26.25)** | **0.0500 (15.60)** | **0.4210 (15.51)** |
| | | SelectiveNet | 0.9246 (57.67) | 1.5540 (58.10) | 0.6266 (97.14) | 2.0810 (67.66) | 4.2466 (57.52) |
| | 70% | TEM | **0.0417 (60.38)** | **0.1076 (75.24)** | **0.0008 (33.02)** | **0.0505 (25.61)** | **0.5040 (32.18)** |
| | | SelectiveNet | 1.3013 (73.65) | 2.1454 (75.39) | 0.6083 (93.70) | 2.1041 (68.53) | 5.5219 (78.88) |
| | 90% | TEM | **0.0417 (88.96)** | **0.1079 (93.97)** | **0.0008 (54.18)** | **0.0558 (46.40)** | **0.6313 (66.90)** |
| | | SelectiveNet | 1.5963 (90.38) | 2.6392 (94.30) | 0.6394 (99.95) | 2.6975 (89.78) | 6.9872 (87.01) |
| | Original | - | 0.0438 | 0.1273 | 0.0016 | 0.0549 | 0.8391 |
| | 10% | TEM | **0.0308 (16.80)** | **0.1069 (29.04)** | **0.0014 (61.17)** | **0.0416 (6.03)** | **0.5222 (5.79)** |
| | | SelectiveNet | 0.9464 (53.09) | 1.2297 (45.05) | 0.3200 (49.72) | 2.0732 (55.11) | 3.2588 (52.76) |
| | 30% | TEM | **0.0447 (34.79)** | **0.1069 (29.04)** | **0.0014 (61.17)** | **0.0434 (16.12)** | **0.4281 (9.03)** |
| | | SelectiveNet | 1.0185 (57.71) | 1.2813 (47.15) | 0.3517 (51.15) | 1.9370 (52.73) | 3.2699 (52.42) |
| TimesNet | 50% | TEM | **0.0455 (71.70)** | **0.1281 (60.91)** | **0.0014 (61.17)** | **0.0451 (32.45)** | **0.4413 (13.66)** |
| | | SelectiveNet | 0.8966 (54.48) | 0.5844 (25.75) | 0.3560 (54.01) | 1.8190 (49.63) | 3.5668 (57.96) |
| | 70% | TEM | **0.0455 (71.70)** | **0.1281 (82.69)** | **0.0014 (65.83)** | **0.0485 (46.81)** | **0.4270 (16.90)** |
| | | SelectiveNet | 1.2080 (70.16) | 1.7400 (79.76) | 0.5239 (78.78) | 3.4055 (85.40) | 4.7244 (78.43) |
| | 90% | TEM | **0.0459 (84.58)** | **0.1276 (94.16)** | **0.0014 (65.83)** | **0.0502 (64.89)** | **0.6104 (37.96)** |
| | | SelectiveNet | 1.5088 (90.83) | 2.8587 (92.41) | 0.6379 (97.73) | 3.3017 (90.90) | 5.8539 (93.66) |

model achieves target coverage. If no model achieves target coverage, then we calculate the percentage error reduction and difference between target coverage and actual coverage. The model with the lowest sum of prediction error and coverage difference is considered the best performing model for that target coverage, dataset, model configuration.

TEM *reduces prediction error across all five models by between* $11.1 - 39.0\%$ *on average for target coverages* $\phi(g) < 50\%$. The highest prediction error reduction was achieved using target coverages $\phi(g) \in \{10\%, 30\%\}$, where TEM achieves on average $21.0\%$ reduction in prediction error across all models and datasets using the Aggregated energy inference method.

The largest error reduction was achieved with the Informer model. For target coverages $\phi(g) \in \{10\%, 30\%\}$, TEM reduces Informer's prediction error by over $34.1\%$. TEM selective forecasting also significantly reduces error for the best performing baseline deterministic models, achieving up to $45.5\%$ reduction for PatchTST and $49.1\%$ for TimesNet for target coverages $\phi(g) \in \{10\%, 30\%\}$.

The average actual coverage $\phi(g)$ recorded for target coverages $\phi(g) \in \{10\%, 30\%\}$ is $21.4\%$ and $32.7\%$ respectively across all models and datasets. This result is expected, as Informer has the lowest deterministic prediction accuracy among all 5 tested deterministic baseline models and has the most room for error reduction.

TEM does not always achieve the target coverage, with actual coverage being on average $11.0\%$ and $17.4\%$ lower than the target coverage for target coverages $\phi(g) \in \{70\%, 90\%\}$ respectively. Most of this difference comes from the PatchTST and TimesNet models, which have the lowest deterministic prediction error among all tested models. We have also noted that the Energy optimization inference method achieves on average $3.5\%$ lower actual coverage than the Aggregated energy inference method across all tested target coverages, but provides up to $30.4\%$ lower prediction error (see Appendix A.1.1 for a more detailed comparison).

## 5.2 Comparison against selected EBM baselines

Configurations of TEM were compared against baseline EBM models. As seen in Table 1, TEM compares favorably against the EBM model baselines. The baseline EB-NARX model *has the overall second highest deterministic prediction error of all the tested models and TEM configurations.* EB-NARX only outperforms the Informer model on all datasets, having on average $51.6\%$ lower error. EB-NARX on average has a $407.1\%$ higher error than the state-of-the-art deterministic TimesNet and PatchTST models across all datasets However, EB-NARX has the lowest deterministic prediction error for the Weather dataset, outperforming both of the state-of-the-art TimesNet and PatchTST models by $50.0\%$ and $27.3\%$ respectively. TEM outperforms EB-NARX on all datasets using selective forecasting, having up to $87.0\%$ lower error for lower target coverages $\phi(g) \in \{10\%, 30\%\}$

The observed EB-NARX performance is expected and can be attributed to the fact that EB-NARX is originally a regression model which tends to suffer from error accumulation when predicting for longer forecast horizons. Furthermore, EB-NARX generates over 2000 samples and performs energy minimization with gradient descent for each time-step $t$, causing inference to be over 5 times slower than TEM.

## 5.3 Comparison against SelectiveNet baselines

Configurations of TEM were compared against baseline SelectiveNet models, which are based on the only other end-to-end deep learning framework that enables selectively rejecting predictions based on user-defined target coverage. We adapted SelectiveNet for the time-series forecasting task as described in Section 4.1. Three architectures of the adapted SelectiveNet were trained for each of the state-of-the-art forecasting models. 6 coverages $c \in \{10\%, 30\%, 50\%, 70\%, 90\%, 100\%\}$ were selected to train SelectiveNet models, which contain the coverages used in the original SelectiveNet paper (Geifman & El-Yaniv, 2019) and $10\%$ and $30\%$ to evaluate performance for lower target coverages. As with prior experiments, 3 SelectiveNet models were trained per architecture, initialized with random seeds, to reduce the likelihood for non-representative results.

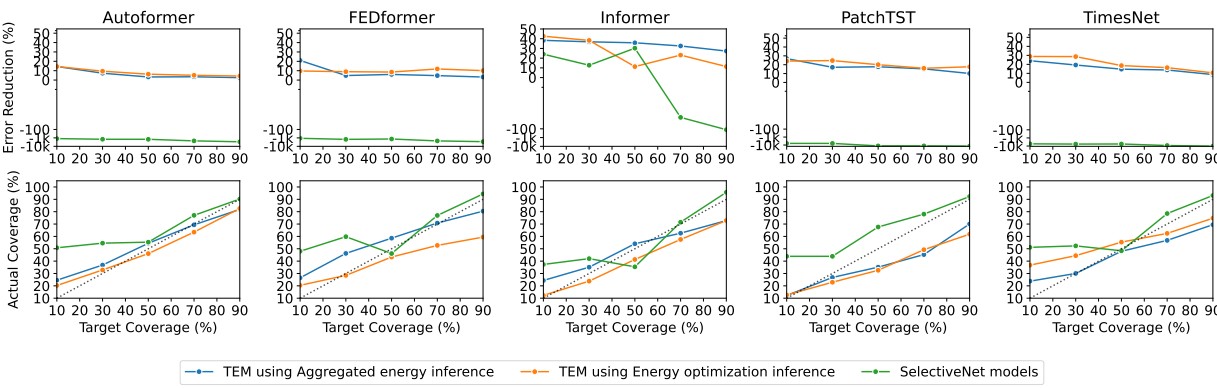

Figure 4: Prediction error reduction (figures at the top), target and actual coverage percentages (figures at the bottom) for TEM and SelectiveNet, across selected target coverages $c \in \{10\%, 30\%, 50\%, 70\%, 90\%\}$ on models Autoformer, FEDformer, Informer, PatchTST, and TimesNet. The top figures' Y-axes use log scale to visualize the several orders of magnitude difference in performance between selective forecasting with TEM and SelectiveNet. For bottom figures, the dotted line represents the ideal case, where actual coverage is equal to target coverage.

As seen in Table 1, SelectiveNet failed to consistently decrease prediction error, across all target coverages, for all tested models across all datasets. TEM using the Aggregated energy inference method outperformed SelectiveNet in every case in terms of selective risk, where TEM reduces error by on average 16.5% across all models and datasets and SelectiveNet increases error by 4244.3% However, SelectiveNet did achieve target coverage more consistently and had on average 14.0% higher coverage than TEM across all tested target coverages, which can be explained by the way SelectiveNet models are trained (see more details in Appendix A.2.2). Furthermore, SelectiveNet did manage to reduce prediction error for the Informer models, achieving on average 18.4% lower error for target coverages $\phi(g) \in \{10\%, 30\%\}$ and 30.2% lower error for target coverages $\phi(g) \in \{50\%\}$. Notably, unlike TEM, SelectiveNet actual coverage does not rise monotonically as target coverage increases, but instead has a saw-tooth pattern, where actual coverage for lower target coverages is higher than for higher target coverages. For example, SelectiveNet with Informer achieves on average 42.0% actual coverage for target coverages $\phi(g) \in \{30\%\}$, but only 35.4% actual coverage for target coverage $\phi(g) \in \{50\%\}$. This is expected, as SelectiveNet models are trained for each target coverage independently, meaning that the coverage achieved for one target coverage is not indicative of the coverage that would be achieved for a different chosen target coverage. This makes SelectiveNet less predictable than TEM in terms of how changing target coverage will affect actual selective coverage and selective risk.

## 5.4 Ablation study and additional experiments

To evaluate the impact that different components of TEM have on selective forecasting performance, we conducted an ablation study and additional experiments. We conducted an ablation study on the two proposed TEM inference methods, Aggregated energy and Energy optimization, comparing them to each other as well as to a naive baseline where energy $E_\theta(X, \hat{Y})$ was directly used to estimate uncertainty (see more details in Appendix A.1.1). Results show that the Aggregated energy inference method achieves on average 10.9% lower selective risk than the Energy optimization inference method, but has 3.5% lower coverage. However, both methods outperform the naive baseline method by 461.9% and 462.2% respectively, showing that both proposed inference methods are superior to the naive baseline. We also conducted an ablation study on the proposed joint training method, comparing it to a configuration where only self-supervised Contrastive Divergence learning was used to train TEM models (see more details in Appendix A.1.2). Results show that TEM without joint training increases forecasting error for selective forecasting by up to 2798.6% across

all coverages, making it unsuitable for practical use. Furthermore, we conducted additional experiments to evaluate TEM performance for univariate time-series forecasting (see more details in Appendix A.2.1). TEM manages to reduce forecast error to a similar degree as for the multivariate forecasting case suggesting that the effectiveness of TEM is not significantly impacted by the dimensionality of the forecasting task and demonstrating that TEM can be used for selective univariate time-series forecasting. Finally, we conducted additional experiments to further analyze the performance of SelectiveNet and identify potential reasons for its poor performance compared to TEM (see more details in Appendix A.2.2). Experiments show that SelectiveNet models tend to converge to a stable coverage level, close to the target coverage. Furthermore, SelectiveNet models are overly conservative during training, selecting too many forecasts and achieving higher coverage than desired, resulting in higher prediction error.

## 6 Conclusions and Future Work

This paper proposes the Time-Energy Model, an energy-based model framework for time-series forecasting. TEM addresses challenges of applying energy-based models to time-series forecasting by providing a framework to parameterize, train, and perform inference with EBMs on time-series data using deep neural networks. TEM introduces selective forecasting that enables the EBM to estimate model confidence, allowing the end-user to selectively reject predictions based on potential forecast error. TEM is parameterized and trained using a proposed joint training method that improves existing baseline EBM models by having a lower inference latency and significantly higher prediction accuracy. As shown in the experiments on 5 state-of-the-art forecasting models TEM can improve the prediction accuracy of encoder-decoder deterministic time-series forecasting models. Experiments show that TEM increases the prediction accuracy over known state-of-the-art forecasting models by up to 49.1% on 5 benchmark datasets. Also, TEM has 87.0% lower error than baseline EBM models and 4244.3% lower error than SelectiveNet models, while also providing significantly faster inference than the former.

In future work, we will extend the TEM framework and develop improved architectures and inference methods that provide better error reduction while maintaining higher coverage. Furthermore, we will apply TEM for time-series outlier and anomaly detection. Finally, we will explore the use of TEM with time-series foundation models.

## Acknowledgements

This research was partially funded by the 6G-XCEL project under the Horizon Europe programme and the H2020 FEVER project (Grant Agreement No. 864537) within the European Union's research and innovation initiatives.

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

# A Appendix

## A.1 Ablation study details

In this section, we provide detailed results for the ablation study conducted to evaluate the impact that proposed TEM inference methods and TEM joint training have on selective forecasting performance.

### A.1.1 Comparison of TEM inference method performance

In this section, we provide detailed results for selective forecasting using both TEM inference methods: Energy optimization inference and Aggregated energy inference. As shown in Table 3, experiments indicate that both Aggregated energy and Energy optimization are effective and reduce prediction error by more than 16.4%. For target coverages $\phi(g) \in \{10\%, 30\%\}$, the Energy optimization method had around 11.0% lower selective coverage but achieved 8.2% lower prediction error than the Aggregated energy method. However, for target coverages $\phi(g) \in \{50\%, 70\%\}$, the Energy optimization method had 9.2% lower selective coverage while achieving 7.9% *higher* prediction error. On average, across all target coverages, the Energy optimization method yielded around 6.2% lower selective coverage and 1.9% lower prediction error than the Aggregated energy method. However, this means that Energy optimization tends not to achieve target coverages as often as the Aggregated energy method. Energy optimization is therefore *recommended for applications where lower prediction error is prioritized over higher selective coverage*. Notably, it is also possible to use a combination of both inference methods, depending on end-user requirements or desired error bounds, as the use of either method does not require retraining the TEM model.

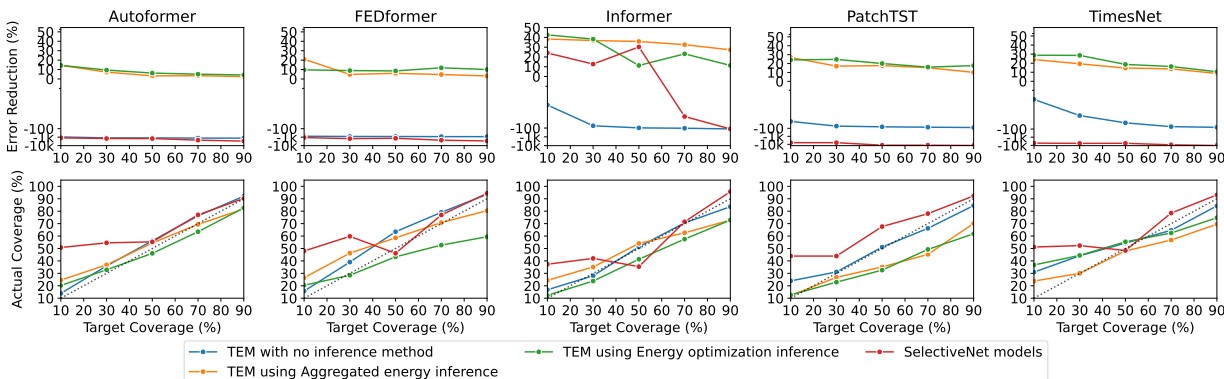

Figure 5: Prediction error reduction (figures at the top), target and actual coverage percentages (figures at the bottom) for TEM models using Aggregated energy and Energy optimization inference methods as well as the naive baseline using $E_\theta(X, \hat{Y})$ directly for estimating uncertainty and selecting forecasts across selected target coverages $\phi(g) \in \{10\%, 30\%, 50\%, 70\%, 90\%, 100\%\}$ on models Autoformer, FEDformer, Informer, PatchTST, and TimesNet. The top figures' Y-axes use log scale to visualize the several orders of magnitude difference in performance between selective forecasting with TEM and SelectiveNet. For bottom figures, the dotted line represents the ideal case, where actual coverage is equal to target coverage.

We also evaluate the performance of TEM without using either Aggregated energy or Energy optimization inference methods. In this case, we use the energy value at the model's output, $E_\theta(X, \hat{Y})$, directly for estimating model uncertainty and selecting forecasts. As shown in Figure 5, the experiments show without using either of the proposed inference methods TEM fails to reduce forecast error and instead increases it by on average 445.6% across all coverages. This is a 461.9% difference when compared to the Aggregated energy inference method and a 462.2% difference when compared to the Energy optimization inference method. However, the results also show that using $E_\theta(X, \hat{Y})$ directly for selecting forecasts yields slightly higher coverage than using either Aggregated energy or Energy optimization inference methods, on average 7.8% higher coverage than Aggregated energy inference method and 14.9% higher coverage than Energy

Table 2: TEM performance comparison across Aggregated energy and Energy optimization inference methods. Results show selective risk and empirical coverage (in parentheses) for target coverages $\phi(g) \in 10\%, 30\%, 50\%, 70\%, 90\%$. Best performing models for specific target coverages are marked **bold**.

| Model | | Dataset | | | | |
|---|---|---|---|---|---|---|
| | | **ETTh1** | **ETTh2** | **Weather** | **Exchange Rate** | **National Illness** |
| **Energy-based model EB-NARX** | | 0.2154 | 0.3003 | **0.0008** | 0.697 | 4.0934 |
| **TEM Autoformer with Aggregated Energy Inference** | Original | 0.0876 | 0.1577 | 0.0079 | 0.0899 | 1.1758 |
| | 10 % | 0.0906 (16.80) | 0.1591 (28.39) | 0.0076 (57.13) | **0.0712 (17.60)** | **0.5634 (2.78)** |
| | 30 % | 0.0889 (51.79) | 0.1591 (28.39) | 0.0076 (57.13) | **0.0708 (26.73)** | **1.0141 (20.14)** |
| | 50 % | 0.0889 (51.79) | 0.1568 (59.22) | 0.0076 (57.13) | **0.0902 (61.93)** | **1.0128 (42.13)** |
| | 70 % | 0.0871 (78.64) | 0.1568 (59.22) | 0.0075 (73.91) | **0.0897 (69.35)** | **1.0464 (65.74)** |
| | 90 % | 0.0881 (87.45) | 0.1568 (76.78) | 0.0075 (73.91) | **0.0890 (87.96)** | **1.0974 (83.10)** |
| **TEM FEDformer with Aggregated Energy Inference** | Original | 0.0772 | 0.1184 | 0.011 | 0.0653 | 1.0503 |
| | 10 % | **0.0782 (45.86)** | **0.0958 (17.75)** | 0.0104 (61.27) | **0.0354 (3.19)** | **0.6754 (4.40)** |
| | 30 % | **0.0782 (45.86)** | **0.1192 (75.68)** | 0.0104 (61.27) | 0.0616 (34.74) | **0.8927 (13.66)** |
| | 50 % | **0.0770 (94.77)** | **0.1192 (75.68)** | 0.0104 (61.27) | 0.0616 (34.74) | **0.8470 (26.39)** |
| | 70 % | **0.0770 (94.77)** | **0.1192 (75.68)** | 0.0105 (67.49) | 0.0634 (72.38) | **0.8748 (43.75)** |
| | 90 % | **0.0770 (94.77)** | **0.1190 (85.77)** | 0.0105 (67.49) | 0.0655 (91.33) | **0.9229 (62.73)** |
| **TEM Informer with Aggregated Energy Inference** | Original | 0.6461 | 1.1877 | 0.3313 | 0.73 | 4.6609 |
| | 10 % | 0.6008 (37.08) | 1.2057 (13.05) | **0.0046 (54.75)** | 0.1828 (7.81) | **4.0733 (9.03)** |
| | 30 % | 0.6008 (37.08) | 1.1754 (27.66) | **0.0046 (54.75)** | 0.2000 (8.11) | 4.4375 (47.92) |
| | 50 % | 0.6460 (89.32) | 1.1845 (49.45) | **0.0046 (54.75)** | 0.1921 (10.47) | 4.3605 (66.20) |
| | 70 % | 0.6460 (89.32) | 1.1814 (70.75) | **0.0046 (59.67)** | 0.2818 (12.98) | 4.5612 (80.32) |
| | 90 % | 0.6438 (93.87) | 1.1850 (85.72) | **0.0046 (59.67)** | 0.4741 (34.02) | 4.5576 (90.74) |
| **TEM PatchTST with Aggregated Energy Inference** | Original | **0.0416** | **0.1079** | 0.0011 | 0.0617 | **0.7324** |
| | 10 % | 0.0340 (15.32) | 0.0735 (11.98) | **0.0008 (21.02)** | 0.0488 (6.33) | **0.4686 (6.25)** |
| | 30 % | 0.0413 (30.05) | 0.1077 (62.31) | **0.0008 (21.02)** | 0.0526 (7.60) | **0.4209 (12.96)** |
| | 50 % | 0.0418 (55.94) | 0.1077 (62.31) | **0.0008 (26.25)** | 0.0500 (15.60) | **0.4210 (15.51)** |
| | 70 % | 0.0417 (60.38) | 0.1076 (75.24) | **0.0008 (33.02)** | 0.0505 (25.61) | **0.5040 (32.18)** |
| | 90 % | 0.0417 (88.96) | 0.1079 (93.97) | **0.0008 (54.18)** | 0.0558 (46.40) | **0.6313 (66.90)** |
| **TEM TimesNet with Aggregated Energy Inference** | Original | 0.0438 | 0.1273 | 0.0016 | **0.0549** | 0.8391 |
| | 10 % | **0.0308 (16.80)** | 0.1069 (29.04) | 0.0014 (61.17) | **0.0416 (6.03)** | 0.5222 (5.79) |
| | 30 % | **0.0447 (34.79)** | 0.1069 (29.04) | 0.0014 (61.17) | **0.0434 (16.12)** | 0.4281 (9.03) |
| | 50 % | **0.0455 (71.70)** | 0.1281 (60.91) | 0.0014 (61.17) | **0.0451 (32.45)** | 0.4413 (13.66) |
| | 70 % | **0.0455 (71.70)** | 0.1281 (82.69) | 0.0014 (65.83) | **0.0485 (46.81)** | 0.4270 (16.90) |
| | 90 % | **0.0459 (84.58)** | 0.1276 (94.16) | 0.0014 (65.83) | **0.0502 (64.89)** | 0.6104 (37.96) |
| **TEM Autoformer with Energy Optimization Inference** | Original | 0.0876 | 0.1577 | 0.0079 | 0.0899 | 1.1758 |
| | 10 % | **0.0870 (23.03)** | **0.1295 (6.77)** | **0.0072 (49.13)** | 0.0770 (14.91) | 0.8180 (7.41) |
| | 30 % | **0.0877 (36.83)** | **0.1438 (33.00)** | **0.0072 (49.13)** | 0.0823 (27.69) | 0.9259 (17.59) |
| | 50 % | **0.0868 (50.32)** | **0.1560 (55.28)** | **0.0072 (53.79)** | 0.0848 (44.15) | 1.0024 (26.85) |
| | 70 % | **0.0877 (70.49)** | **0.1523 (66.87)** | **0.0072 (72.44)** | 0.0881 (66.12) | 1.0443 (41.44) |
| | 90 % | **0.0874 (89.62)** | **0.1582 (93.41)** | **0.0073 (92.71)** | 0.0890 (80.94) | 1.0252 (56.48) |
| **TEM FEDformer with Energy Optimization Inference** | Original | 0.0772 | 0.1184 | 0.011 | 0.0653 | 1.0503 |
| | 10 % | 0.0837 (19.13) | 0.1195 (1.33) | **0.0104 (59.76)** | 0.0590 (14.18) | 0.6012 (7.64) |
| | 30 % | 0.0829 (31.89) | 0.1191 (16.16) | **0.0104 (59.76)** | 0.0624 (26.84) | 0.6012 (7.64) |
| | 50 % | 0.0802 (53.28) | 0.1172 (46.53) | **0.0104 (59.76)** | 0.0644 (44.40) | 0.6421 (12.50) |
| | 70 % | 0.0721 (63.05) | 0.1172 (46.53) | **0.0104 (72.11)** | 0.0616 (65.46) | 0.6245 (16.44) |
| | 90 % | 0.0721 (63.05) | 0.1172 (46.53) | **0.0106 (88.18)** | 0.0632 (69.51) | 0.6732 (29.86) |
| **TEM Informer with Energy Optimization Inference** | Original | 0.6461 | 1.1877 | 0.3313 | 0.73 | 4.6609 |
| | 10 % | **0.3781 (4.70)** | **1.0603 (15.52)** | 0.0277 (24.73) | **0.3151 (3.87)** | 4.0905 (12.50) |
| | 30 % | **0.6071 (28.61)** | **1.0449 (27.14)** | 0.0277 (24.73) | **0.2251 (7.81)** | 4.0831 (30.79) |
| | 50 % | **0.6502 (48.61)** | **1.2784 (47.18)** | 0.0277 (24.73) | **0.8804 (32.40)** | 4.9517 (53.70) |
| | 70 % | **0.6382 (77.61)** | **1.1949 (74.68)** | 0.0262 (25.95) | **0.5810 (42.69)** | 4.5389 (66.67) |
| | 90 % | **0.6463 (91.31)** | **1.1938 (93.81)** | 0.1971 (42.59) | **0.6350 (55.94)** | 4.4865 (80.56) |
| **TEM PatchTST with Energy Optimization Inference** | Original | **0.0416** | **0.1079** | 0.0011 | 0.0617 | **0.7324** |
| | 10 % | **0.0423 (12.14)** | **0.0928 (23.16)** | 0.0007 (10.86) | **0.0444 (5.69)** | 0.4118 (11.11) |
| | 30 % | **0.0415 (28.87)** | **0.0916 (48.95)** | 0.0007 (10.86) | **0.0448 (9.18)** | 0.4084 (16.67) |
| | 50 % | **0.0419 (53.24)** | **0.0918 (55.66)** | 0.0007 (10.86) | **0.0533 (22.40)** | 0.4692 (20.83) |
| | 70 % | **0.0415 (90.83)** | **0.1103 (74.94)** | 0.0007 (10.86) | **0.0574 (44.31)** | 0.4503 (25.23) |
| | 90 % | **0.0416 (98.28)** | **0.1082 (93.90)** | 0.0006 (15.13) | **0.0584 (65.96)** | 0.4554 (35.65) |
| **TEM TimesNet with Energy Optimization Inference** | Original | 0.0438 | 0.1273 | 0.0016 | **0.0549** | 0.8391 |
| | 10 % | 0.0459 (84.58) | **0.0797 (9.60)** | **0.0012 (56.03)** | 0.0456 (13.68) | **0.5387 (4.40)** |
| | 30 % | 0.0459 (84.58) | **0.0745 (31.04)** | **0.0012 (56.03)** | 0.0527 (22.34) | **0.4703 (12.73)** |
| | 50 % | 0.0459 (84.58) | 0.1290 (73.26) | **0.0012 (56.03)** | 0.0525 (32.01) | **0.4489 (15.51)** |
| | 70 % | 0.0459 (84.58) | 0.1290 (73.26) | 0.0013 (71.81) | 0.0526 (44.44) | **0.4722 (22.69)** |
| | 90 % | 0.0459 (84.58) | 0.1278 (90.36) | 0.0015 (93.78) | 0.0530 (54.96) | **0.5636 (34.49)** |

Table 3: TEM performance comparison across Aggregated energy (Agg. Energy) and Energy optimization (Energy Opt.) inference methods. Results show selective risk and empirical coverage (in parentheses) for target coverages $\phi(g) \in \{10\%, 30\%, 50\%, 70\%, 90\%\}$. Best performing models for specific target coverages are marked **bold**.

| Base Model | Coverage | Method | Dataset | | | | |
|---|---|---|---|---|---|---|---|
| | | | ETTh1 | ETTh2 | Weather | Exchange Rate | National Illness |
| **EB-NARX** | Original | - | 0.2154 | 0.3003 | **0.0008** | 0.697 | 4.0934 |
| **Autoformer** | Original | - | 0.0876 | 0.1577 | 0.0079 | 0.0899 | 1.1758 |
| | 10% | Agg. Energy | 0.0906 (16.80) | **0.1591 (28.39)** | 0.0076 (57.13) | **0.0712 (17.60)** | **0.5634 (2.78)** |
| | | Energy Opt. | **0.0870 (23.03)** | 0.1295 (6.77) | **0.0072 (49.13)** | 0.0770 (14.91) | 0.8180 (7.41) |
| | 30% | Agg. Energy | 0.0889 (51.79) | 0.1591 (28.39) | 0.0076 (57.13) | **0.0708 (26.73)** | 1.0141 (20.14) |
| | | Energy Opt. | **0.0877 (36.83)** | **0.1438 (33.00)** | **0.0072 (49.13)** | 0.0823 (27.69) | **0.9259 (17.59)** |
| | 50% | Agg. Energy | 0.0889 (51.79) | 0.1568 (59.22) | 0.0076 (57.13) | 0.0902 (61.93) | 1.0128 (42.13) |
| | | Energy Opt. | **0.0868 (50.32)** | **0.1560 (55.28)** | **0.0072 (53.79)** | 0.0848 (44.15) | 1.0024 (26.85) |
| | 70% | Agg. Energy | 0.0871 (78.64) | 0.1568 (59.22) | 0.0075 (73.91) | **0.0897 (69.35)** | 1.0464 (65.74) |
| | | Energy Opt. | **0.0877 (70.49)** | **0.1523 (66.87)** | **0.0072 (72.44)** | 0.0881 (66.12) | 1.0443 (41.44) |
| | 90% | Agg. Energy | 0.0881 (87.45) | 0.1568 (76.78) | 0.0075 (73.91) | 0.0890 (87.96) | 1.0974 (83.10) |
| | | Energy Opt. | **0.0874 (89.62)** | **0.1582 (93.41)** | **0.0073 (92.71)** | 0.0890 (80.94) | 1.0252 (56.48) |
| **FEDformer** | Original | - | 0.0772 | 0.1184 | 0.011 | 0.0653 | 1.0503 |
| | 10% | Agg. Energy | **0.0782 (45.86)** | **0.0958 (17.75)** | **0.0104 (61.27)** | **0.0354 (3.19)** | 0.6754 (4.40) |
| | | Energy Opt. | 0.0837 (19.13) | 0.1195 (1.33) | 0.0104 (59.76) | 0.0590 (14.18) | **0.6012 (7.64)** |
| | 30% | Agg. Energy | **0.0782 (45.86)** | 0.1192 (75.68) | 0.0104 (61.27) | **0.0616 (34.74)** | 0.8927 (13.66) |
| | | Energy Opt. | 0.0829 (31.89) | 0.1191 (16.16) | **0.0104 (59.76)** | 0.0624 (26.84) | **0.6012 (7.64)** |
| | 50% | Agg. Energy | **0.0770 (94.77)** | **0.1192 (75.68)** | 0.0104 (61.27) | 0.0616 (34.74) | 0.8470 (26.39) |
| | | Energy Opt. | 0.0802 (53.28) | 0.1172 (46.53) | **0.0104 (59.76)** | 0.0644 (44.40) | 0.6421 (12.50) |
| | 70% | Agg. Energy | **0.0770 (94.77)** | **0.1192 (75.68)** | 0.0105 (67.49) | 0.0634 (72.38) | 0.8748 (43.75) |
| | | Energy Opt. | 0.0721 (63.05) | 0.1172 (46.53) | **0.0104 (72.11)** | 0.0616 (65.46) | 0.6245 (16.44) |
| | 90% | Agg. Energy | **0.0770 (94.77)** | **0.1190 (85.77)** | 0.0105 (67.49) | 0.0655 (91.33) | 0.9229 (62.73) |
| | | Energy Opt. | 0.0721 (63.05) | 0.1172 (46.53) | **0.0106 (88.18)** | 0.0632 (69.51) | 0.6732 (29.86) |
| **Informer** | Original | - | 0.6461 | 1.1877 | 0.3313 | 0.73 | 4.6609 |
| | 10% | Agg. Energy | **0.6008 (37.08)** | 1.2057 (13.05) | **0.0046 (54.75)** | **0.1828 (7.81)** | 4.0733 (9.03) |
| | | Energy Opt. | 0.3781 (4.70) | **1.0603 (15.52)** | 0.0277 (24.73) | 0.3151 (3.87) | **4.0905 (12.50)** |
| | 30% | Agg. Energy | **0.6008 (37.08)** | 1.1754 (27.66) | **0.0046 (54.75)** | **0.2000 (8.11)** | 4.4375 (47.92) |
| | | Energy Opt. | 0.6071 (28.61) | **1.0449 (27.14)** | 0.0277 (24.73) | 0.2251 (7.81) | **4.0831 (30.79)** |
| | 50% | Agg. Energy | 0.6460 (89.32) | 1.1845 (49.45) | **0.0046 (54.75)** | **0.1921 (10.47)** | 4.3605 (66.20) |
| | | Energy Opt. | 0.6502 (48.61) | 1.2784 (47.18) | 0.0277 (24.73) | 0.8804 (32.40) | 4.9517 (53.70) |
| | 70% | Agg. Energy | 0.6460 (89.32) | **1.1814 (70.75)** | **0.0046 (59.67)** | **0.2818 (12.98)** | 4.5612 (80.32) |
| | | Energy Opt. | **0.6382 (77.61)** | 1.1949 (74.68) | 0.0262 (25.95) | 0.5810 (42.69) | 4.5389 (66.67) |
| | 90% | Agg. Energy | 0.6438 (93.87) | **1.1850 (85.72)** | 0.0046 (59.67) | 0.4741 (34.02) | 4.5576 (90.74) |
| | | Energy Opt. | 0.6463 (91.31) | 1.1938 (93.81) | 0.1971 (42.59) | 0.6350 (55.94) | 4.4865 (80.56) |
| **PatchTST** | Original | - | **0.0416** | **0.1079** | 0.0011 | 0.0617 | **0.7324** |
| | 10% | Agg. Energy | **0.0340 (15.32)** | **0.0735 (11.98)** | 0.0008 (21.02) | **0.0488 (6.33)** | 0.4686 (6.25) |
| | | Energy Opt. | 0.0423 (12.14) | 0.0928 (23.16) | **0.0007 (10.86)** | 0.0444 (5.69) | **0.4118 (11.11)** |
| | 30% | Agg. Energy | **0.0413 (30.05)** | 0.1077 (62.31) | **0.0008 (21.02)** | 0.0526 (7.60) | 0.4209 (12.96) |
| | | Energy Opt. | 0.0415 (28.87) | **0.0916 (48.95)** | 0.0007 (10.86) | **0.0448 (9.18)** | **0.4084 (16.67)** |
| | 50% | Agg. Energy | 0.0418 (55.94) | 0.1077 (62.31) | **0.0008 (26.25)** | 0.0500 (15.60) | 0.4210 (15.51) |
| | | Energy Opt. | 0.0419 (53.24) | **0.0918 (55.66)** | 0.0007 (10.86) | **0.0533 (22.40)** | 0.4692 (20.83) |
| | 70% | Agg. Energy | 0.0417 (60.38) | **0.1076 (75.24)** | **0.0008 (33.02)** | 0.0505 (25.61) | **0.5040 (32.18)** |
| | | Energy Opt. | **0.0415 (90.83)** | 0.1103 (74.94) | 0.0007 (10.86) | **0.0574 (44.31)** | 0.4503 (25.23) |
| | 90% | Agg. Energy | 0.0417 (88.96) | **0.1079 (93.97)** | 0.0008 (54.18) | 0.0558 (46.40) | 0.6313 (66.90) |
| | | Energy Opt. | **0.0416 (98.28)** | 0.1082 (93.90) | 0.0006 (15.13) | **0.0584 (65.96)** | 0.4554 (35.65) |
| **TimesNet** | Original | - | 0.0438 | 0.1273 | 0.0016 | **0.0549** | 0.8391 |
| | 10% | Agg. Energy | **0.0308 (16.80)** | **0.1069 (29.04)** | 0.0014 (61.17) | 0.0416 (6.03) | **0.5222 (5.79)** |
| | | Energy Opt. | 0.0459 (84.58) | 0.0797 (9.60) | **0.0012 (56.03)** | 0.0456 (13.68) | 0.5387 (4.40) |
| | 30% | Agg. Energy | **0.0447 (34.79)** | 0.1069 (29.04) | 0.0014 (61.17) | 0.0434 (16.12) | 0.4281 (9.03) |
| | | Energy Opt. | 0.0459 (84.58) | **0.0745 (31.04)** | **0.0012 (56.03)** | 0.0527 (22.34) | **0.4703 (12.73)** |
| | 50% | Agg. Energy | 0.0455 (71.70) | 0.1281 (60.91) | 0.0014 (61.17) | **0.0451 (32.45)** | 0.4413 (13.66) |
| | | Energy Opt. | 0.0459 (84.58) | 0.1290 (73.26) | **0.0012 (56.03)** | 0.0525 (32.01) | **0.4489 (15.51)** |
| | 70% | Agg. Energy | **0.0455 (71.70)** | **0.1281 (82.69)** | 0.0014 (65.83) | **0.0485 (46.81)** | 0.4270 (16.90) |
| | | Energy Opt. | 0.0459 (84.58) | 0.1290 (73.26) | **0.0013 (71.81)** | 0.0526 (44.44) | **0.4722 (22.69)** |
| | 90% | Agg. Energy | **0.0459 (84.58)** | **0.1276 (94.16)** | 0.0014 (65.83) | **0.0502 (64.89)** | 0.6104 (37.96) |
| | | Energy Opt. | 0.0459 (84.58) | 0.1278 (90.36) | **0.0015 (93.78)** | 0.0530 (54.96) | 0.5636 (34.49) |

Table 4: Relative change in forecasting error for deterministic models using joint training and using Constrastive Divergence only. Positive percentages indicate the relative forecasting error increase when models using only Contrastive Divergence (without joint training), **negative percentages** indicate the relative forecasting error reduction.

| Deterministic model | ETTh1 | ETTh2 | Weather | Exchange Rate | National Illness |
|---|---|---|---|---|---|
| **FEDformer** | +364.51% | +306.25% | +3085.45% | +495.10% | +118.19% |
| **Autoformer** | +501.37% | +273.18% | +5396.20% | +379.64% | +85.07% |
| **Informer** | +212.24% | +81.71% | **-1.51%** | +276.10% | +116.32% |
| **PatchTST** | +53.37% | +111.86% | +18.18% | +71.47% | +172.57% |
| **TimesNet** | +55.48% | +85.55% | **-18.75%** | +101.28% | +119.22% |

optimization inference method. These results show that *sampling a single energy value at the model's output* $E_\theta(X, \hat{Y})$ *does not provide enough information for estimating model uncertainty and selecting forecasts.*

Sampling energy values from around the model's output (as done when using Aggregated energy and Energy optimization inference methods) provides more accurate model uncertainty estimates. Intuitively, if the model is confident in its prediction, the energy value is low, then the energy values around the model's output should also be low – model should be confident in forecasting very similar values. If the energy values on and around the forecast are high, then that indicates that the energy surface around the forecast is not low and the model might have not seen many similar examples in its training data.

However, aggregating the energy values can disrupt the natural ordering of the energy values (as generally, low energy values indicate high probability and high model confidence). We have observed that it when performing the calibration and ranking of energy values, the natural ordering of energy values is generally not disrupted, but instead it improves the accuracy of the estimation of mean prediction error $\epsilon$ for each energy interval (as shown in Figure 3). More accurate prediction error estimates enable TEM selective forecasting to more consistently select forecasts with lower prediction error, reducing prediction error more for the same target coverages.

### A.1.2 TEM performance when trained only using self-supervised learning

In this section, we provide the results of experiments training deterministic and TEM models without joint training, only using Contrastive Divergence self-supervised learning. The key difference between TEM joint training and training using only Contrastive Divergence is in how the model parameters are optimized. With joint training (Section 3.3), the deterministic model $A_\Psi$ is first trained using supervised learning to directly minimize forecasting error, and then the energy-based model parameters $\theta$ are trained using Contrastive Divergence while keeping $A_\Psi$ parameters frozen. In contrast, when training using only Contrastive Divergence, all model parameters (both $A_\Psi$ and $E_\theta$) are trained simultaneously using self-supervised learning to learn the entire data distribution, without directly optimizing for forecasting accuracy.

As shown in Table 4, the experiments indicate that without joint training, deterministic models have on average 498.4% higher forecasting error across all 5 models and 5 datasets. However, on the Weather dataset, using Contrastive Divergence yielded a slight increase in forecasting accuracy for models TimesNet and Informer, 18.8% and 1.5% respectively. The performance reduction is particularly significant for Autoformer and FEDformer models, where training using only Contrastive Divergence results in up to 5396.2% and 3085.5% higher error on average across all datasets. These results show that *the proposed TEM joint training method is essential for maintaining high deterministic forecasting accuracy.*

As a result of poor deterministic performance, TEM models without joint training do not yield any relative error reduction when compared to TEM trained with joint training. As seen in Figure 6, on average, TEM without joint training increases forecasting error when using selective forecasting by 2798.6% across all coverages. Notably, TEM without joint training yields 89.5% lower forecasting error than SelectiveNet models. However, neither TEM without joint training nor SelectiveNet models yield sufficient deterministic performance to be used effectively in practice. Poor deterministic performance of TEM models trained without joint training can be attributed to the models not being trained using a loss function that is directly

optimized for forecasting error. EBMs are trained using Contrastive Divergence, which is a loss function designed to learn the entire data distribution, not directly optimize forecasting error. Without the utilization of supervised learning, as proposed in this work, TEM models (or generative models in general) often cannot show comparable forecasting performance to conventional discriminative deterministic models (Bond-Taylor et al., 2022; Zheng et al., 2023).

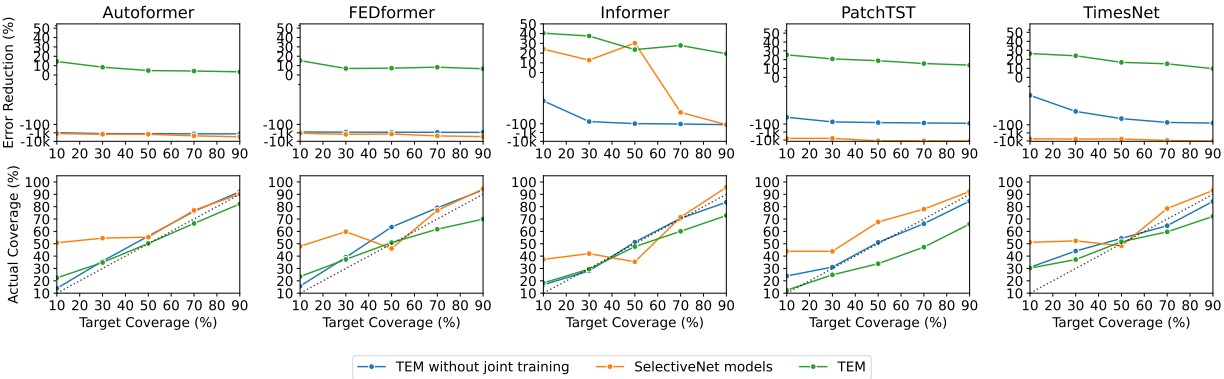

Figure 6: Prediction error reduction (figures at the top), target and actual coverage percentages (figures at the bottom) for TEM models trained with and without joint training, and SelectiveNet for multivariate selective forecasting, across selected target coverages $\phi(g) \in \{10\%, 30\%, 50\%, 70\%, 90\%, 100\%\}$ on models Autoformer, FEDformer, Informer, PatchTST, and TimesNet. The top figures' Y-axes use log scale to visualize the several orders of magnitude difference in performance between selective forecasting with TEM and SelectiveNet. For bottom figures, the dotted line represents the ideal case, where actual coverage is equal to target coverage.

## A.2   Additional experiments

In this section, we provide detailed results for additional experiments evaluating TEM performance for selective *univariate* time-series forecasting and further analysis of SelectiveNet performance.

### A.2.1   TEM performance for univariate selective forecasting

In this section, we evaluate TEM performance for univariate selective forecasting. In these experiments, TEM was trained to forecast using only observed data for one feature (the target variable) TEM achieves similar performance in the univariate forecasting scenario to multivariate forecasting. As seen in Table 5, across all models and datasets, TEM achieves an average prediction error reduction of 23.6% for target coverages $\phi(g) \in \{10\%, 30\%\}$, which is within 2% of the multivariate case. This suggests that the effectiveness of TEM is not significantly impacted by the dimensionality of the forecasting task.

As can be seen in Figure 7, the performance gap between TEM Aggregated energy and Energy optimization inference methods is notably smaller for univariate selective forecasting. The Energy optimization method achieves on average 10.9% lower prediction error than Aggregated energy, while having 3.5% lower coverage. SelectiveNet shows improved performance in the univariate case, achieving 80.7% lower prediction error compared to SelectiveNet applied for multivariate forecasting across all models and datasets. This could be attributed to the fact that univariate forecasting is a comparatively easier task, as the forecasting model does not need to consider the interactions between covariate features. SelectiveNet, like in the multivariate selective forecasting case, also performs well with the Informer architecture, in some cases reducing error by up to 95% and outperforming TEM with Aggregated energy in most scenarios across all five datasets. However, SelectiveNet still on average increases forecasting error by 820.0% compared to deterministic models, and performs 4944.5% worse than TEM across all target coverages and across all models and datasets, showing that it does not generalize across different model architectures and datasets well, unlike TEM.

Table 5: TEM performance comparison for *univariate selective forecasting*. Results show selective risk and empirical coverage (in parentheses) for target coverages $\phi(g) \in \{10\%, 30\%, 50\%, 70\%, 90\%\}$. Best performing models for specific target coverages are marked **bold**.

| Base Model | Coverage | Method | Dataset | | | | |
|---|---|---|---|---|---|---|---|
| | | | ETTh1 | ETTh2 | Weather | Exchange Rate | National Illness |
| **EB-NARX** | | | 0.2154 | 0.3003 | **0.0008** | 0.697 | 4.0934 |
| **Autoformer** | Original | - | 0.0876 | 0.1578 | 0.0078 | 0.0897 | 1.1766 |
| | 10% | TEM | **0.0869 (58.18)** | **0.1471 (12.09)** | 0.0076 (47.58) | **0.0535 (7.17)** | **0.7639 (7.41)** |
| | | SelectiveNet | 0.1667 (53.82) | 2.8711 (51.16) | 0.0071 (56.58) | 1.8717 (49.61) | 3.4556 (50.35) |
| | 30% | TEM | **0.0869 (58.18)** | **0.1537 (52.55)** | **0.0076 (47.58)** | **0.0866 (67.01)** | **0.9514 (25.23)** |
| | | SelectiveNet | 0.1116 (55.25) | 1.2875 (52.96) | 0.0412 (49.22) | 1.8666 (58.53) | 3.8514 (50.45) |
| | 50% | TEM | **0.0869 (58.18)** | **0.1537 (52.55)** | **0.0075 (53.51)** | **0.0866 (67.01)** | **1.1224 (72.45)** |
| | | SelectiveNet | 0.1392 (54.56) | 1.5872 (48.52) | 0.0163 (54.23) | 1.6051 (50.08) | 4.0139 (58.30) |
| | 70% | TEM | **0.0864 (90.90)** | **0.1564 (72.45)** | **0.0075 (70.76)** | **0.0866 (67.01)** | **1.1224 (72.45)** |
| | | SelectiveNet | 0.1384 (64.39) | 3.7973 (70.33) | 0.1271 (80.03) | 1.9077 (78.92) | 4.1406 (70.55) |
| | 90% | TEM | **0.0864 (90.90)** | **0.1578 (87.71)** | **0.0075 (92.00)** | **0.0889 (91.73)** | **1.1416 (86.57)** |
| | | SelectiveNet | 1.4219 (93.44) | 4.2827 (96.40) | 0.0404 (98.96) | 3.1112 (89.03) | 4.7521 (84.59) |
| **FEDformer** | Original | - | 0.0772 | 0.1185 | 0.011 | 0.0648 | 1.0498 |
| | 10% | TEM | **0.0594 (18.63)** | **0.0960 (20.39)** | **0.0102 (62.40)** | **0.0444 (8.24)** | **0.5466 (3.70)** |
| | | SelectiveNet | 0.1176 (49.65) | 0.0007 (0.41) | 0.1456 (55.01) | 0.1360 (50.32) | 1.5589 (65.82) |
| | 30% | TEM | **0.0773 (63.75)** | **0.1209 (64.70)** | **0.0102 (62.40)** | **0.0551 (42.14)** | **0.7346 (22.22)** |
| | | SelectiveNet | 0.1002 (53.32) | 0.1260 (75.43) | 0.0066 (49.75) | 0.1068 (46.51) | 1.3187 (55.67) |
| | 50% | TEM | **0.0773 (63.75)** | **0.1209 (64.70)** | **0.0102 (62.40)** | **0.0619 (74.50)** | **0.7089 (29.86)** |
| | | SelectiveNet | 0.1084 (53.59) | 0.1410 (84.12) | 0.0071 (52.80) | 0.1029 (53.64) | 1.2956 (53.68) |
| | 70% | TEM | **0.0769 (69.77)** | **0.1200 (74.55)** | **0.0103 (68.51)** | **0.0619 (74.50)** | **0.7250 (38.66)** |
| | | SelectiveNet | 0.2000 (88.23) | 0.1698 (97.25) | 0.0108 (65.73) | 0.1538 (68.20) | 2.7279 (69.05) |
| | 90% | TEM | **0.0775 (91.93)** | **0.1184 (92.26)** | **0.0103 (68.51)** | **0.0654 (93.60)** | **0.9505 (66.90)** |
| | | SelectiveNet | 0.2103 (91.04) | 0.1329 (82.71) | 0.0076 (98.14) | 0.2258 (87.39) | 3.8039 (88.10) |
| **Informer** | Original | - | 0.6459 | 1.1883 | 0.331 | 0.7299 | 4.6562 |
| | 10% | TEM | **0.6140 (14.03)** | **1.1228 (16.51)** | **0.0079 (58.72)** | **0.2263 (6.63)** | **3.8635 (24.54)** |
| | | SelectiveNet | 0.2162 (75.14) | 0.0791 (34.24) | 0.0026 (22.52) | 0.1179 (60.32) | 1.6542 (45.53) |
| | 30% | TEM | **0.5962 (38.37)** | **1.1450 (32.89)** | **0.0079 (58.72)** | **0.2263 (6.63)** | **3.8635 (24.54)** |
| | | SelectiveNet | 0.1080 (27.82) | 0.0682 (41.66) | 0.0064 (46.22) | 0.1346 (55.71) | 2.6890 (63.18) |
| | 50% | TEM | **0.6437 (90.94)** | **1.1450 (32.89)** | **0.0079 (58.72)** | **0.4293 (18.12)** | **4.3947 (39.12)** |
| | | SelectiveNet | 0.2090 (83.82) | 0.0581 (29.83) | 0.0120 (61.19) | 0.1204 (32.81) | 1.5793 (42.86) |
| | 70% | TEM | **0.6437 (90.94)** | **1.1749 (54.26)** | **0.0069 (64.92)** | **0.4847 (35.66)** | **4.7776 (67.36)** |
| | | SelectiveNet | 0.3951 (95.74) | 0.1736 (99.35) | 0.0077 (97.83) | 0.1917 (63.12) | 0.4581 (12.22) |
| | 90% | TEM | **0.6437 (90.94)** | **1.1766 (79.21)** | **0.0069 (64.92)** | **0.5828 (59.65)** | **4.5910 (90.51)** |
| | | SelectiveNet | 0.1277 (92.33) | 0.1537 (64.71) | 0.0179 (98.68) | - | 2.0234 (51.53) |
| **PatchTST** | Original | - | **0.0416** | **0.1075** | 0.0011 | 0.0617 | **0.7324** |
| | 10% | TEM | **0.0309 (9.13)** | **0.0817 (30.45)** | **0.0006 (36.06)** | **0.0383 (2.87)** | **0.5478 (6.25)** |
| | | SelectiveNet | 0.9696 (51.24) | 0.5454 (61.69) | 0.0034 (51.76) | 1.5066 (45.68) | 3.3755 (50.57) |
| | 30% | TEM | **0.0408 (39.34)** | **0.0817 (30.45)** | **0.0006 (36.06)** | **0.0440 (11.98)** | **0.6111 (11.34)** |
| | | SelectiveNet | 0.9696 (51.24) | 0.5454 (61.69) | 0.0034 (51.76) | 1.5066 (45.68) | 3.3755 (50.57) |
| | 50% | TEM | **0.0405 (56.43)** | **0.1085 (63.04)** | **0.0007 (47.94)** | **0.0529 (34.11)** | **0.5964 (20.60)** |
| | | SelectiveNet | 0.9696 (51.24) | 0.6768 (70.70) | 0.0070 (58.87) | 1.5066 (45.68) | 3.3755 (50.57) |
| | 70% | TEM | **0.0404 (63.89)** | **0.1088 (74.40)** | **0.0009 (81.80)** | **0.0536 (41.87)** | **0.6068 (38.19)** |
| | | SelectiveNet | 1.3973 (80.32) | 0.7841 (78.95) | 0.0069 (86.33) | 2.7152 (74.00) | 4.5747 (72.65) |
| | 90% | TEM | **0.0412 (87.25)** | **0.1079 (92.44)** | **0.0009 (89.11)** | **0.0570 (73.25)** | **0.6580 (62.73)** |
| | | SelectiveNet | 1.5713 (92.21) | 1.1102 (90.96) | 0.0059 (95.16) | 3.0774 (89.21) | 6.2271 (89.40) |
| **TimesNet** | Original | - | 0.0438 | 0.1273 | 0.0016 | **0.0549** | 0.8391 |
| | 10% | TEM | **0.0308 (16.80)** | **0.1069 (29.04)** | **0.0014 (61.17)** | **0.0406 (7.97)** | **0.5222 (5.79)** |
| | | SelectiveNet | 0.9464 (53.09) | 1.2297 (45.05) | 0.3200 (49.72) | 2.0732 (55.11) | 3.2588 (52.76) |
| | 30% | TEM | **0.0447 (34.79)** | **0.1069 (29.04)** | **0.0014 (61.17)** | **0.0436 (25.59)** | **0.4281 (9.03)** |
| | | SelectiveNet | 1.0185 (57.71) | 1.2813 (47.15) | 0.3517 (51.15) | 1.9370 (52.73) | 3.2699 (52.94) |
| | 50% | TEM | **0.0455 (71.70)** | **0.1281 (60.91)** | **0.0014 (61.17)** | **0.0451 (32.45)** | **0.4413 (13.66)** |
| | | SelectiveNet | 0.8966 (54.48) | 0.5844 (25.75) | 0.3560 (54.01) | 1.8190 (49.63) | 3.5668 (57.96) |
| | 70% | TEM | **0.0455 (71.70)** | **0.1281 (82.69)** | **0.0014 (65.83)** | **0.0483 (49.70)** | **0.5914 (31.48)** |
| | | SelectiveNet | 1.2080 (70.16) | 1.7400 (79.76) | 0.5239 (78.78) | 3.4055 (85.40) | 4.7244 (78.43) |
| | 90% | TEM | **0.0459 (84.58)** | **0.1276 (94.16)** | **0.0014 (65.83)** | **0.0513 (72.50)** | **0.6939 (61.57)** |
| | | SelectiveNet | 1.5088 (90.83) | 2.8587 (92.41) | 0.6379 (97.73) | 3.3017 (90.90) | 5.8539 (93.66) |

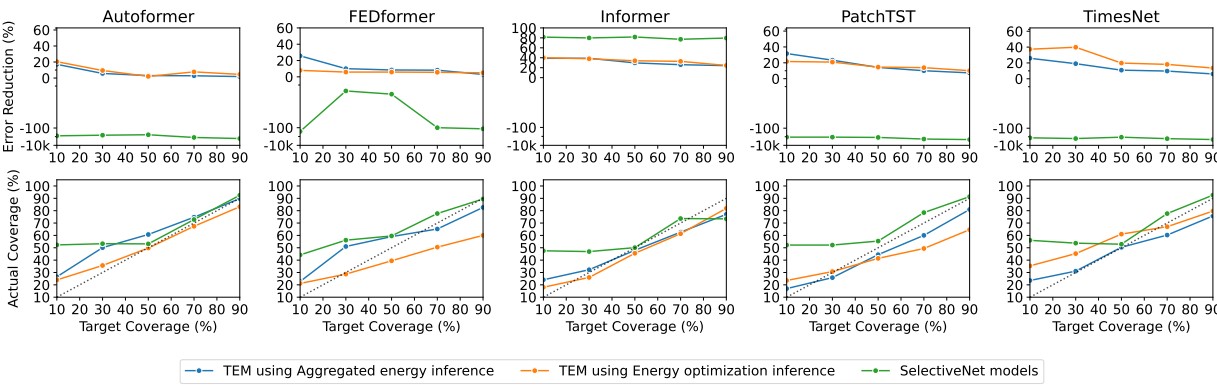

Figure 7: Prediction error reduction (figures at the top), target and actual coverage percentages (figures at the bottom) for TEM and SelectiveNet for *univariate selective forecasting*, across selected target coverages $\phi(g) \in \{10\%, 30\%, 50\%, 70\%, 90\%, 100\%\}$ on models Autoformer, FEDformer, Informer, PatchTST, and TimesNet. The top figures' Y-axes use log scale to visualize the several orders of magnitude difference in performance between selective forecasting with TEM and SelectiveNet. For bottom figures, the dotted line represents the ideal case, where actual coverage is equal to target coverage.

### A.2.2   SelectiveNet performance analysis

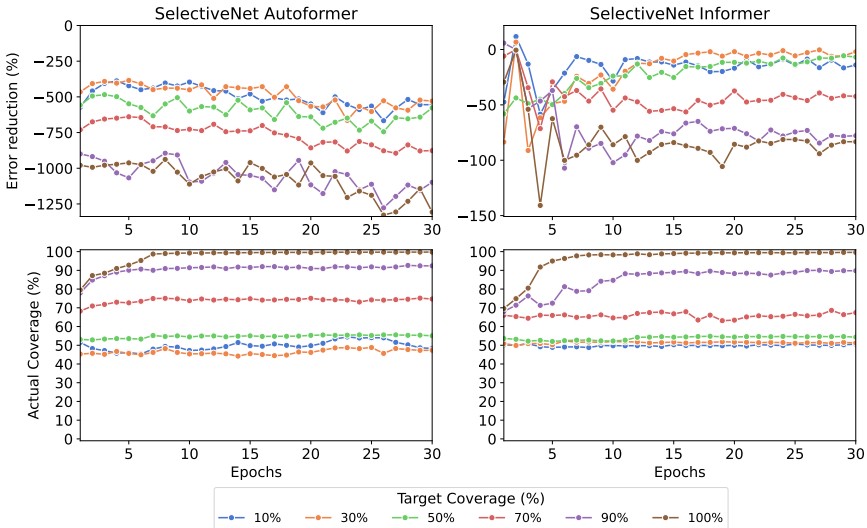

Figure 8: Error reduction (figures at the top) and actual coverage percentages (figures at the bottom) for SelectiveNet Autoformer and Informer models on the ETTh2 dataset for epochs $\in [1, 30]$ and target coverages $\phi(g) \in \{10\%, 30\%, 50\%, 70\%, 90\%, 100\%\}$.

In this section we provide additional analysis on the performance of SelectiveNet to identify the reasons for its poor performance compared to TEM. We trained 3 SelectiveNet models using different seeds for each of the Autoformer and Informer variants of SelectiveNet on the ETTh2 dataset for epochs $\in [1, 30]$ and target coverages $\phi(g) \in \{10\%, 30\%, 50\%, 70\%, 90\%, 100\%\}$.

As shown in Figure 8, SelectiveNet models tend to converge to a stable coverage, as the actual coverage percentages reach close to the target coverage after a few epochs. As the coverage converges, the forecasting performance of SelectiveNet also stops improving despite further training. This indicates that the loss

function used by SelectiveNet prioritizes achieving the target coverage during optimization, rather than minimizing prediction error, which is consistent with all prior experimental results. And since the models converge, even if training were extended with more epochs, the performance would not meaningfully improve.

Furthermore, we notice that for lower coverages, SelectiveNet consistently achieves significantly higher actual coverage than target coverage. On average, SelectiveNet achieves 35.0% and 15.5% higher coverage across both models for coverages $\phi(g) \in \{10\%, 30\%\}$, respectively. This shows that SelectiveNet is overly conservative during training, selecting too many forecasts and achieving higher coverage than desired and, as a result, higher prediction error. These patterns are consistent across all tested models and datasets.

We conducted additional experiments to evaluate the performance of SelectiveNet when trained using a traditional loss function – Mean Squared Error, without using the selection head. This SelectiveNet without using the selection head achieved the same deterministic forecasting performance as the base model $A_\Psi$.

### A.3   Additional information on datasets

In this section, we provide more information for the five datasets used in experiments for evaluating TEM performance. The Features column in Table 6 represents the number of features in the dataset, including the target variable. The Dataset Size column in Table 6 shows the number of data points in each of the training, validation, and test subsets.

Table 6: Statistics for Time Series datasets used in experiments

| Name | Features | Dataset Size | Frequency | Domain |
|---|---|---|---|---|
| ETTh1 | 7 | (8545, 2881, 2881) | Hourly | Electricity |
| ETTh2 | 7 | (8545, 2881, 2881) | Hourly | Electricity |
| Exchange | 8 | (5120, 665, 1422) | Daily | Exchange Rate |
| Weather | 21 | (36792, 5271, 10540) | Daily | Weather |
| National Illness | 7 | (617, 74, 170) | Weekly | Illness |

### A.4   Contrastive Divergence training

In this section, we provide the joint training algorithm for training TEM with Contrastive Divergence enabling selective forecasting.

---

**Algorithm 1** Calculating Contrastive Divergence (CD) loss

---

**INPUT:**
$E_\theta$ – Energy-based model (EBM)
$X$ – Ground-truth input
$Y^{(0)}$ – Ground-truth output given input $X$
$\eta$ – CD step size
$\alpha_{CD}$ – CD regularizer coefficient
$N_{CD}$ – CD step count
**OUTPUT:**
$\mathcal{L}_{CD}$ - Contrastive Divergence (CD) loss

    $Y^{(1)} \leftarrow \mathcal{N}(0, \sigma^2 I)$
    **for** $i \leftarrow 1$ to $N_{CD}$ **do**
        $\omega \leftarrow \mathcal{N}(0, \sigma^2 I)$
        $Y^{(1)} \leftarrow Y^{(1)} - \eta \nabla_{Y^{(1)}} E_\theta(X, Y^{(1)}) + \omega$                            ▷ Eq. 11
    **end for**
    $E^+ \leftarrow E_\theta(X, Y^{(0)})$
    $E^- \leftarrow E_\theta(X, Y^{(1)})$
    $\mathcal{L}_{CD} \leftarrow (E^+ - E^-) + \alpha_{CD}((E^+)^2 + (E^-)^2)$               ▷ Eq. 12
    **return** $\mathcal{L}_{CD}$

---

## A.5 Additional implementation details on SelectiveNet

In this work, we adapt the SelectiveNet framework for time-series forecasting by modifying its architecture and loss function to work with state-of-the-art deterministic forecasting models. Similar to TEM, our SelectiveNet implementation reuses architecture from the deterministic forecasting model $A_\Psi$ to maintain the model's ability to capture temporal dependencies and patterns in the input time series (as described in Section 3.2) The adapted SelectiveNet architecture adds three heads on top of the deterministic model $A_\Psi$: 1) a selection head is added on top of the encoder of $A_\Psi$, that outputs selection scores (and is used to implement the selective function $g$ as shown in Equation 6), 2) a prediction head is appended to the decoder of the $A_\Psi$ and produces deterministic forecasts, and 3) an auxiliary prediction head is added to the decoder of the $A_\Psi$ and also produces predictions, as described in the original SelectiveNet paper (Geifman & El-Yaniv, 2019).

## A.6 Additional implementation details on TEM

In this section, we provide additional implementation details. The base implementation of TEM models presented in this paper is available at `https://github.com/JonasBrusokas/Time-Energy-Model`. The deterministic forecasting models (Informer, Autoformer, FEDformer, PatchTST, and TimesNet) were trained according to the hyperparameters specified in their respective papers (Zhou et al., 2021; Wu et al., 2021; Zhou et al., 2022; Wu et al., 2023; Nie et al., 2023). The model implementations were adapted from an open-source time-series forecasting model repository that implements state-of-the-art models and runs experiments with the same hyperparameters, ensuring consistent and reproducible model architectures and training procedures. This repository is available at `https://github.com/thuml/Time-Series-Library`.

For all experiments we use the same training, validation, and test data splits as the original papers. Sequence length $m$ was set to 96 for all models, with prediction horizon $h$ set to 48.

For deterministic forecasting models $A_\Psi$, we recreate the model architectures and training procedures as described in the original papers, using the same hyperparameters. All models were trained with 2-layer encoders, 1-layer decoders. All deterministic models were trained using the Mean Squared Error loss function, using Adam optimizer with learning rate 0.0001 and dropout rate of 0.05. Deterministic models were trained for up to 30 epochs, using early stopping with patience parameter of 3. Transformer-based models (Informer, Autoformer, FEDformer, and PatchTST) were trained with 8 attention heads and 512 dimensionality embedding, attention, and feed-forward layers.

## A.7 Standard deviations across experiments

As mentioned in Section 5.1, the results reported are the average of 3 runs. We provide the standard deviations across the 3 runs for model and dataset combination in Table 7.

Table 7: Standard deviations for selective risk for the 5 tested models and SelectiveNet models across 3 runs.

| Model | Dataset | | | | |
| --- | --- | --- | --- | --- | --- |
| | ETTh1 | ETTh2 | Weather | Exchange Rate | National Illness |
| TEM Autoformer | 0.0817 | 0.1421 | 0.0074 | 0.0929 | 0.4474 |
| TEM FEDformer | 0.0833 | 0.1041 | 0.0078 | 0.0708 | 0.4442 |
| TEM Informer | 0.3570 | 0.6497 | 0.6855 | 0.6503 | 2.4439 |
| TEM PatchTST | 0.0479 | 0.1021 | 0.0016 | 0.0657 | 0.4128 |
| TEM TimesNet | 0.0536 | 0.1255 | 0.0025 | 0.0582 | 0.4065 |
| SelectiveNet Autoformer | 0.0977 | 0.2146 | 0.0106 | 0.2769 | 0.5654 |
| SelectiveNet FEDformer | 0.1738 | 0.6707 | 0.0467 | 0.1086 | 0.2989 |
| SelectiveNet Informer | 0.4178 | 0.7700 | 0.0243 | 0.4219 | 0.4342 |
| SelectiveNet PatchTST | 0.2400 | 0.5423 | 0.0334 | 0.8309 | 0.5265 |
| SelectiveNet TimesNet | 0.1276 | 0.4620 | 0.2089 | 0.7244 | 0.7938 |

## A.8 Additional experiments with LSTM

Table 8: Comparison of TEM performance with Informer, TimesNet, and LSTM models. Results show selective risk and empirical coverage (in parentheses) for target coverages $\phi(g) \in 10\%, 30\%, 50\%, 70\%, 90\%$. Best performing models for specific target coverages are marked **bold**, second-best performers are underlined.

| Model | | Dataset | | | | |
|---|---|---|---|---|---|---|
| | | **ETTh1** | **ETTh2** | **Weather** | **Exchange Rate** | **National Illness** |
| **TEM Informer with Aggregated Energy Inference** | Original | 0.6461 | 1.1877 | 0.3313 | 0.73 | 4.6609 |
| | 10 % | 0.6008 (37.08) | 1.2057 (13.05) | 0.0046 (54.75) | 0.1828 (7.81) | 4.0733 (9.03) |
| | 30 % | 0.6008 (37.08) | 1.1754 (27.66) | 0.0046 (54.75) | 0.2000 (8.11) | 4.4375 (47.92) |
| | 50 % | 0.6460 (89.32) | 1.1845 (49.45) | 0.0046 (54.75) | 0.1921 (10.47) | 4.3605 (66.20) |
| | 70 % | 0.6460 (89.32) | 1.1814 (70.75) | 0.0046 (59.67) | 0.2818 (12.98) | 4.5612 (80.32) |
| | 90 % | 0.6438 (93.87) | 1.1850 (85.72) | 0.0046 (59.67) | 0.4741 (34.02) | 4.5576 (90.74) |
| **TEM TimesNet with Aggregated Energy Inference** | Original | **0.0438** | 0.1273 | **0.0016** | **0.0549** | **0.8391** |
| | 10 % | **0.0308 (16.80)** | **0.1069 (29.04)** | **0.0014 (61.17)** | **0.0416 (6.03)** | **0.5222 (5.79)** |
| | 30 % | **0.0447 (34.79)** | **0.1069 (29.04)** | **0.0014 (61.17)** | **0.0434 (16.12)** | **0.4281 (9.03)** |
| | 50 % | **0.0455 (71.70)** | 0.1281 (60.91) | **0.0014 (61.17)** | **0.0451 (32.45)** | **0.4413 (13.66)** |
| | 70 % | **0.0455 (71.70)** | **0.1281 (82.69)** | **0.0014 (65.83)** | **0.0485 (46.81)** | **0.4270 (16.90)** |
| | 90 % | **0.0459 (84.58)** | **0.1276 (94.16)** | **0.0014 (65.83)** | **0.0502 (64.89)** | **0.6104 (37.96)** |
| **TEM LSTM with Aggregated Energy Inference** | Original | 0.1514 | **0.1272** | 0.0035 | 0.085 | 4.496 |
| | 10 % | 0.0443 (1.91) | 0.1263 (4.96) | 0.0026 (45.27) | 0.0598 (0.72) | - (0.00) |
| | 30 % | 0.1407 (25.35) | 0.1191 (20.02) | 0.0026 (45.27) | 0.0793 (4.92) | 3.6716 (33.68) |
| | 50 % | 0.1582 (50.00) | **0.1264 (51.92)** | 0.0020 (60.16) | 0.0551 (17.28) | 3.2849 (44.44) |
| | 70 % | 0.1406 (77.44) | 0.1264 (51.92) | 0.0024 (65.77) | 0.0499 (39.28) | 3.9742 (63.89) |
| | 90 % | 0.1406 (77.44) | 0.1264 (51.92) | 0.0034 (98.03) | 0.0677 (64.96) | 4.2693 (85.07) |

In this section, we provide additional experiments with LSTM deterministic models for time-series forecasting. As with the rest of the models, we use the same training, validation, and test data splits as the original papers. Sequence length $m$ was set to 96 for all models, with prediction horizon $h$ set to 48. The architecture selected used 1 layer of LSTM cells with 128 hidden units for the encoder and a fully connected layer with 128 hidden units for the decoder. The LSTM was trained using the Mean Squared Error loss function, using Adam optimizer with learning rate 0.0001 and dropout rate of 0.05. The LSTM was trained for up to 30 epochs, using early stopping with patience parameter of 3. We used the same training procedure to train TEM with LSTM as the deterministic model, as with the rest of the deterministic models.

We provide the results for TEM with LSTM in Table 8, where we compare TEM with LSTM to TEM with Informer and TimesNet, the worst and best performing models in the deterministic forecasting experiments. The deterministic LSTM model numerically underperforms against the top-performing TimesNet model, but outperforms the Informer model in all observed cases. Results indicate that TEM with LSTM achieves, on average, 71.8% lower prediction error than TEM with Informer across all target coverages. TEM is able to reduce the prediction error of the LSTM model by up to 30% across all datasets. However, we observe that TEM with LSTM often yields lower than the target coverage. When target coverages are low ($\phi(g) \in \{10\%, 30\%\}$), the TEM with LSTM model yields up to 10% lower coverage than the target coverage. It is important to note, however, that the encoder of the LSTM model is only 1 layer deep and the model itself is significantly smaller than all the other models tested, which results in a worse performing encoder and less accurate representations of the input time series.

## A.9 Error-bounded selective forecasting with TEM

In this paper we propose TEM for selective forecasting based on target coverage, but the framework can be extended to use other selection criteria. This section describes how TEM can be used to reject predictions based on estimated prediction error bounds.

The key difference from the original coverage-based approach is that instead of selecting energy intervals to achieve a target coverage, we can use the relationship between energy values $E_\theta(X, \hat{Y})$ and prediction errors $\epsilon(E_\theta(X, \hat{Y}))$ to reject predictions that are likely to have errors above a specified threshold. Specifically, after partitioning the energy range into intervals $E_{\theta i}$ and calculating mean prediction error $\epsilon$ for each interval (as described in Section 3.4), we can define selective forecasting with error bounds as:

$$(A_\Psi, E_\theta)(X, \varepsilon) \triangleq \begin{cases} A_\Psi(X) = \hat{Y} & \text{if } \epsilon(E_\theta(X, \hat{Y})) \leq \varepsilon, \\ \text{None}, & \text{otherwise}, \end{cases} \tag{15}$$

where $\varepsilon$ is the maximum acceptable prediction error and $\epsilon(E_\theta(X, \hat{Y}))$ is the estimated prediction error for energy $E_\theta(X, \hat{Y})$.

This approach allows end-users to specify the maximum error bound for their use-case. Like with the coverage-based approach, this approach also allows dynamic adjustment of selection criteria based on the utility of the prediction and potential penalty for errors.

Like the original coverage-based approach, the error bound-based selection can use either the Aggregated energy or Energy optimization inference methods described in Section 3.4. Both the coverage-based and error bound-based approaches can be used interchangeably without retraining the model, as they both utilize the same underlying relationship between energy values and prediction errors.

### A.10 Comparison of selective forecasting with probabilistic forecasting models

In this section, we provide a comparison of TEM with a probabilistic forecasting model – TimeGrad.

The TimeGrad model is an autoregressive denoising diffusion model designed for multivariate probabilistic time series forecasting (Rasul et al., 2021). Uses recurrent neural networks (RNNs) to encode past sequences and temporal dependencies using the hidden state. Inference with TimeGrad is performed using annealed Langevin dynamics, which generates multiple samples to obtain empirical quantiles of uncertainty for each prediction.

To enable a fair comparison between TimeGrad (a probabilistic model) and TEM, we adapt TimeGrad for selective forecasting as follows: First, we generate multiple samples from TimeGrad and use the variance between these samples as an uncertainty measure, (rather than using energy values as in TEM). We then apply a similar selective forecasting procedure as described in Section 3.4:

1. Divide predictions into intervals based on their variance

2. Calculate prediction error $\epsilon$ and coverage for each interval

3. Select forecasts that meet the target coverage criteria

This adaptation allows us to directly compare TimeGrad's selective forecasting capabilities with TEM. Note, that we do not perform any type of aggregation or additional sampling (beyond generating 10 forecasts) as we do in TEM selective forecasting. For TimeGrad, we also do not rank the intervals by prediction error, instead assuming that the variance between samples is a good proxy for prediction error.

Experiments are conducted using the openly available GitHub repository from the authors TimeGrad (Rasul et al., 2021). TimeGrad is trained for 20 epochs, with a batch size of 64 and GRU is used to model the hidden state. Other hyperparameters are the same hyperparameters as in the original paper or the default values used in the repository. Inference is performed autoregressively, by predicting one time-step at a time. We use annealed Langevin dynamics to generate $S = 10$ forecasts from TimeGrad. The same datasets are used in experiments and same preprocessing is applied to the data as in prior experiments. For selective forecasting, we use the same target coverages $\phi(g) \in \{10\%, 30\%, 50\%, 70\%, 90\%\}$ as in prior experiments.

As can be seen in Table 9, TEM with TimesNet outperforms TimeGrad in all cases. The best performing TEM TimesNet model has over 91% lower error across all datasets for the deterministic forecasting case and can consistently further reduce error by performing selective forecasting. TimeGrad is more competitive against Informer, outperforming it on the ETTh1, ETTh2, and Exchange Rate datasets. However, Informer has, on average, 42% lower error than TimeGrad, because of the very high errors recorded on the National Illness and Weather datasets.

Table 9: Comparison of TEM performance with Informer, TimesNet, and TimeGrad models. Results show selective risk and empirical coverage (in parentheses) for target coverages $\phi(g) \in 10\%, 30\%, 50\%, 70\%, 90\%$. Best performing models for specific target coverages are marked **bold**, second-best performers are underlined.

| Model | | Dataset | | | | |
|---|---|---|---|---|---|---|
| | | **ETTh1** | **ETTh2** | **Weather** | **Exchange Rate** | **National Illness** |
| **TEM Informer with Aggregated Energy Inference** | Original | 0.6461 | 1.1877 | 0.3313 | 0.73 | 4.6609 |
| | 10 % | 0.6008 (37.08) | 1.2057 (13.05) | 0.0046 (54.75) | 0.1828 (7.81) | 4.0733 (9.03) |
| | 30 % | 0.6008 (37.08) | 1.1754 (27.66) | 0.0046 (54.75) | 0.2000 (8.11) | 4.4375 (47.92) |
| | 50 % | 0.6460 (89.32) | 1.1845 (49.45) | 0.0046 (54.75) | 0.1921 (10.47) | 4.3605 (66.20) |
| | 70 % | 0.6460 (89.32) | 1.1814 (70.75) | 0.0046 (59.67) | 0.2818 (12.98) | 4.5612 (80.32) |
| | 90 % | 0.6438 (93.87) | 1.1850 (85.72) | 0.0046 (59.67) | 0.4741 (34.02) | 4.5576 (90.74) |
| **TEM TimesNet with Aggregated Energy Inference** | Original | **0.0438** | **0.1273** | **0.0016** | **0.0549** | **0.8391** |
| | 10 % | **0.0308 (16.80)** | **0.1069 (29.04)** | **0.0014 (61.17)** | **0.0416 (6.03)** | **0.5222 (5.79)** |
| | 30 % | **0.0447 (34.79)** | **0.1069 (29.04)** | **0.0014 (61.17)** | **0.0434 (16.12)** | **0.4281 (9.03)** |
| | 50 % | **0.0455 (71.70)** | **0.1281 (60.91)** | **0.0014 (61.17)** | **0.0451 (32.45)** | **0.4413 (13.66)** |
| | 70 % | **0.0455 (71.70)** | **0.1281 (82.69)** | **0.0014 (65.83)** | **0.0485 (46.81)** | **0.4270 (16.90)** |
| | 90 % | **0.0459 (84.58)** | **0.1276 (94.16)** | **0.0014 (65.83)** | **0.0502 (64.89)** | **0.6104 (37.96)** |
| **TimeGrad** | Original | 0.4235 | 1.1547 | 0.7733 | 0.1459 | 9.1967 |
| | 10 % | 0.3030 (61.95) | 1.3082 (86.67) | 0.8338 (72.53) | - (0.00) | 5.5429 (7.92) |
| | 30 % | 0.4197 (99.13) | 1.3082 (86.67) | 0.7929 (86.70) | 0.1147 (15.35) | 5.5429 (7.92) |
| | 50 % | 0.4224 (99.76) | 1.3082 (86.67) | 0.7929 (86.70) | 0.1201 (32.35) | 5.5429 (7.92) |
| | 70 % | 0.4235 (100.00) | 1.3082 (86.67) | 0.7929 (86.70) | 0.1283 (47.06) | 5.5429 (7.92) |
| | 90 % | 0.4235 (100.00) | 1.3082 (86.67) | 0.7929 (86.70) | 0.1254 (42.46) | 5.5429 (7.92) |

These results can be explained by the fact that TimeGrad is a probabilistic model and is not trained or tuned to perform very accurate deterministic forecasting. As a result, even though TimeGrad can be used for selective forecasting to reduce forecasting error by trading off coverage, it is still not as accurate as TEM. Furthermore, the TimeGrad model is significantly slower during inference because of its autoregressive forecasting and sampling process. Finally, TimeGrad often cannot effectively perform selective forecasting, because the variance between samples is not a consistently good proxy for prediction error. As a result, selective forecasting sometimes increases the forecasting error (as in the ETTh2 and Weather dataset cases).

## A.11 Discussion on applications of TEM to forecasting of multiple time-series

Currently, TEM has been evaluated only for the case of multivariate time-series forecasting where the forecasted time-series is univariate. TEM in principle allows for selective forecasting of multivariate time-series, but this has not been investigated in the context of multivariate time-series forecasting. In particular, when forecasting multiple target variables simultaneously, the relationship between energy values and prediction errors may become more complex. It is possible that for high-dimensional multivariate forecasting problems, the output search space can grow too large to sample energy values and provide accurate prediction error estimates. Energy-based models have been shown to work well even in very high-dimensional spaces (e.g. with image or video data) (Du & Mordatch, 2019), but this has not been investigated in the context of multivariate time-series forecasting and will be part of future work.

