# OpenReview forum: "The Time-Energy Model: Selective Time-Series Forecasting Using Energy-Based Models"
_TMLR — Accepted by TMLR_

### Review · Reviewer_58WR · 2024-12-11

**Summary Of Contributions:**

In this paper, the authors propose a selective prediction framework called the Time Energy Model (TEM) for time-series forecasting that uses an energy-based model (EBM) to model the conditional probability of $Y$ given $X$ where $Y$ is the predicted forecast and $X$ is the time series. Section 2 introduces some basic preliminaries and sets up the problem formulation. Section 3 dives into the details of the framework and has separate sub-sections explaining the architecture, training, and forecasting procedures. Section 4 details a set of experiments on five benchmark datasets to test the efficacy of $TEM$ against $SelectiveNet$ - a state of the art selective prediction framework for classification/regression.

**Audience:**

Yes

**Claims And Evidence:**

Yes

**Requested Changes:**

Although the framework is both interesting and novel, the experiments section doesn't seem to make a convincing case of using TEM over just using the deterministic baselines. It could be the case that forecasting on the datasets used in the paper do not benefit much from selective prediction and thus do not do a good job of highlighting the strengths of TEM. With that in mind, it would be difficult to accept this paper unless TEM's utility can be empirically demonstrated where there is a clear advantage of using a selective forecasting framework like TEM over a deterministic model.

**Strengths And Weaknesses:**

**Strengths:**
1. The framework proposed in the paper does a good job of extending selective prediction to time-series data
2. The problem setup, architecture, and forecasting procedure are all clearly explained (Figure 1 does a particularly good job of summarizing all this information)
3. Unlike SelectiveNet, the target coverage can be changed without re-training the network allowing for flexible risk-coverage trade-off during run-time


**Weaknesses:**

Unfortunately, the experimental results don't appear to be very convincing. As I understand from Table 1, there doesn't appear to be too much of a lift using TEM over using the deterministic baseline models thereby negating the need to do selective forecasting in the first place. To make the results more compact and easier to read, I feel that it might be a good idea to compare area under the risk-coverage curve against baseline risk instead of the individual risk for every coverage level. This would give the reader a quick and easy way to compare the selective prediction models against the deterministic baselines. In addition, it might be helpful for the readers to have some more information on how SelectiveNet was adapted to accommodate time-series data.

---

> ### Author Response · Authors · 2025-01-04
> **Author response and clarifications**
>
> Thank you for your thoughtful feedback regarding the experimental results. We would like to address the concerns that the improvements shown by TEM are not convincing, and would like to clarify several important points:
>
> 1. The benchmark datasets used in our experiments have been extensively studied in the time-series forecasting literature. Improvements between successive state-of-the-art models on these datasets typically range from 5-10%. In this context, TEM's demonstrated ability to reduce prediction error by 15-20% on average (and up to 49.1% in some cases) represents a substantial improvement over existing approaches.
>
> 2. The practical significance of these improvements becomes clearer when considering real-world applications where prediction errors carry penalties - for example in electricity trading, energy consumption forecasting, or financial markets. In these domains, the ability to selectively reject inaccurate predictions while maintaining a desired coverage level provides significant value and enable the end-user to make an informed decision about model confidence and potential prediction error.
>
> 3. TEM offers a unique combination of strengths that sets it apart from both deterministic and probabilistic approaches:
>    - Compared to alternative selective and probabilistic approaches, TEM achieves significantly higher deterministic accuracy
>    - Compared to deterministic models, TEM adds the crucial capability to estimate model confidence and enable selective forecasting
>    - TEM achieves this while maintaining model-agnostic compatibility with any encoder-decoder architecture, as demonstrated with 6 different deterministic architectures in the paper
>
> As for SelectiveNet's adaptation to time-series data, we have added detailed information about this in Appendix Section A.5.
>
> If there are any further changes or clarifications needed, please let us know.
>
> Thank you once again for your comments and for your time!

---

### Review · Reviewer_WaXh · 2024-12-18

**Summary Of Contributions:**

Due to the importance of time-series forecasting in real-world applications, various methods have been developed. However, most of recent methods are constrained to deterministic forecasting, which means that they cannot measure the confidence of their prediction, leading to low utilization in real-world scenarios. To overcome this limitation, this paper proposes selective time-series forecasting with a Time-Energy Model (TEM). Compared to existing not only deterministic models but also models for selective prediction, their method has several distinct characteristics that contribute to improved performance. First, they combine time series forecasting models and energy-based models into a single framework with joint parameterization and training. Second, their method can be applied to any time-series forecasting model with encoder-decoder structures. Finally, they introduce a scalable selective forecasting procedure that narrows the search space of time-series data, the size of which extremely increases according to time length. In the experiments, through selective forecasting, TEM achieves up to a 49.1% reduction in prediction error compared to five leading deterministic models. Additionally, TEM demonstrates up to 87.0% lower error than baseline EBM models and significantly outperforms state-of-the-art selective deep learning models.

**Audience:**

No

**Broader Impact Concerns:**

Can you include the broader impact of this paper in a separate section?

**Claims And Evidence:**

No

**Requested Changes:**

Addressing the **Major Issue** outlined in the Strengths and Weaknesses section is essential for this paper to be considered acceptable.

As for the **Minor Issue**, they do not critically weaken the paper but addressing them would enhance its quality. In particular, resolving concerns 4 and 5 could significantly improve the overall impact of the paper.

**Strengths And Weaknesses:**

# Strengths
1. Important Problem
I think this paper proposes an important problem that has not received much attention but has great meaning.
While most research in time-series forecasting focuses on deterministic prediction, this paper aims to address probabilistic forecasting and selective forecasting. In the real world, a probabilistic problem is also important because it requires measuring the confidence of predictions, which helps users determine if they believe the predictions of models. Also, selective forecasting makes the decision process more sysmatically. I believe that although various works for probabilistic forecasting have already been introduced, selective forecasting is a somewhat newly proposed problem in time-series forecasting.

2. Model-Agnostic Methods
It is not a standalone model but one that can be applied agnostically to other existing models. In spite of the fact that their method can be applied to forecasting models with encoder-decoder structures, it is still meaningful in that many forecasting methods employ encoder-decoder structures.

# Weaknesses

## Major Issue

1. As stated in the manuscript, energy $E_\theta$ cannot be directly used for comparison of different $X$, because it is not normalized. To overcome this, the authors propose Aggregated energy inference or Energy optimization inference. I think the detailed descriptions of why these inference techniques can overcome unnormalized characteristics. Also, can you provide the reason for experimental results in the ablation studies where a naive energy method and their proposed energy methods are applied? In other words, I'm wondering why the proposed inference methods outperform the case where energy values are used naively.

2. In general, a small  $E_\theta$  indicates high probability, while a large  $E_\theta$  suggests low probability. In other words, the order of energy also carries important meaning. However, in step 2 of selective forecasting with TEM, all energy intervals are ranked based on forecasting error, which may disrupt the natural ordering of energy values and potentially affect the interpretability of the model’s energy-based predictions.

3. Joint training is one of the key contributions highlighted by the authors. However, the lack of detailed explanations makes it difficult to understand the specific characteristics of their joint training approach. Could you clarify the differences between using joint training and not using it, as well as its application when leveraging pre-trained deterministic forecasting models?

4. The authors highlight that existing SelectiveNet models do not allow predictions to be rejected based on criteria such as estimated prediction error, whereas their approach enables selection or rejection of forecasts based on such criteria. To support this claim, I believe two points need to be addressed: first, a detailed explanation of the procedure when using other criteria, and second, an outline of the unique features or novel modules that make this capability possible.

5. Selective forecasting is inherently a form of probabilistic forecasting. Therefore, I believe this paper should incorporate recent probabilistic forecasting models that use a straightforward inference technique, where multiple predictions are sampled, and their variance is utilized as a measure of confidence (or as the criterion for rejection). [1,2,3]

## Minor Issue

1. Using Equation (11), negative samples are generated. Due to lack of my knowledge in training energy-based models with negative samples, please provide more details of justification for this method. Specifically, Equation 11 results in $Y^{(1)}$ with lower $E$ which means high probabilities. Then, can $Y^{(1)}$ from Langevin dynamics be regarded as positive samples?

2. There are a few typos in the manuscript, such as the use of ? in place of citations and the incorrect notation $\theta xy$, which should be corrected to $\theta_{xy}$.

3. There is no paragraph which refers to Figure 4.

4. I believe it would be beneficial to further propose a method for refining rejected predictions, enabling them to be corrected and ultimately accepted. Although the problem area defined in this paper is constrained to rejecting well. However, I think the extension to correction is necessary for real-world scenarios.

5. Are there any advanced evaluation metrics for selective forecasting? I think the evaluation metrics should measure not only how well models forecast but also the alignment between test forecasting errors and the rank of energy intervals, which means as the rank changes, test forecasting errors monotonically increase or decrease. Table 1 and Figure 4 indirectly show this alignment, but a more direct way to measure the degree of alignment or misalignment is required. Based on the existing literature on selective predictions, concerns 4 and 5 in **Minor Issue** may not necessarily need to be addressed. However, I believe these concerns become significant when considering real-world applications.

[1] Tashiro et al., CSDI: Conditional Score-based Diffusion Models for Probabilistic Time Series Imputation, 2021
[2] Kollovieh et al., Predict, Refine, Synthesize: Self-Guiding Diffusion Models for Probabilistic Time Series Forecasting, 2023
[3] Rasul et al., Autoregressive Denoising Diffusion Models for Multivariate Probabilistic Time Series Forecasting, 2021

---

> ### Author Response · Authors · 2025-01-04
> **Author response and clarifications**
>
> Thank you for your thoughtful feedback and comments. We have addressed the major issues in the revised paper.
>
> - For major issue 1, we have added a short discussion of our intuition why the proposed inference methods outperform the case where energy values are used naively in the last paragraph of Section A.1.1.
> - For major issue 2, we have added a short discussion on why the ranking of energy intervals based on forecasting error may disrupt the natural ordering of energy values in the last paragraph of Section A.1.1.
> - For major issue 3, we have added additional clarification on the differences between using TEM joint training and using only self-supervised training in Section A.2.2.
> - For major issue 4, we have added an additional section in the Appendix (Section A.9) that describes how TEM could be used to select forecasts based on estimated prediction error.
> - For major issue 5, we have added experiments evaluating the performance of TEM against TimeGrad [3], a state-of-the-art probabilistic forecasting model. We have added the results in Section A.10.
>
> - For minor issues, we have tried to find the mistakes and made corrections in the revised version. For minor issue 4 - there could definitely be utility of refining inaccurate predictions. However, this would require the probabilistic model (in our case, the EBM) to be inherently a significantly stronger point predictor, which is not easy to achieve, as shown by the lacking results of both selective and probabilistic baselines. We believe this to be an direction for future research.
> - For minor issue 5 - we are not aware of any tailored metric that would allow us to accurately quantify this alignment. We have improved the presentation, making it easier to read and interpret experiment results in the tables, figures, and text.
>
> If there are any further suggestions, requested changes or clarifications needed, please let us know.
>
> Thank you once again for your comments and for your time!

---

### Review · Reviewer_3U5f · 2024-12-20

**Summary Of Contributions:**

The paper proposes a framework, called Time-Energy Model (TEM), for selective time series forecasting based on coupling possibly pre-trained point predictors, based on encoder-decoder architectures, with Energy Based Models (EBMs). In this framework the EBM acts as a selector model which selects whether a prediction is trustworthy or not based on a target coverage. To achieve this the authors propose an unsupervised training strategy for the EBM and two inference strategies to sample energy values for a prediction. Moreover, they propose a methodology to achieve different target coverages without need for additional model training.

**Audience:**

Yes

**Broader Impact Concerns:**

I don't have concerns regarding the broader impact of the work.

**Claims And Evidence:**

Yes

**Requested Changes:**

- I would like to see a more precise discussion on the models implementation and training (hyperparameter selection for instance). Moreover, if possible, code used to train the models should be released for reproducibility purposes.
- A discussion on the suitability of the approach for multivariate forecasting, with experimental results (see Weaknesses).
- Experiments for non-transformer based architectures would broaden the scope of the proposed approach. Even simple baseline architectures would suffice in my opinion.
- I would like to see some information on the performance of the SelectiveNet predictor when the selection output is ignored (see Questions), to better understand if the bad performance could be partly explained through an ineherently worse predictor with respect to the standard supervisedly trained baseline.
- Table captions should clarify if the reported results are the best run of 3 or the mean, and also the standard deviation across runs should be reported.
- Readability of Table 1 and similars could be improved, for instance by putting the results for each target coverage close together. The caption states that bold results denote best for each target coverage, however it seems more to denote the best selective approach for each model **and** target coverage. If the purpose of the table is to show that TEM is better than others, model and coverage wise, a table in which lines are grouped by model and then by coverage could drastically improve readability and convey better the message. Moreover, it seems that only the selective error is considered, to decide if a model is better than another. This should be stated more clearly in the caption.
- There were minor problems I found while reading the paper, I summarize them in the following:
	- Sec 5.1, Par. 7: It is stated that the average error reduction is between 11.1 and 39% across models, but then that the best average prediction error reduction was 21%. This seems counterintuitive. Does the first phrase refer to average reduction per dataset across models while the second is the overall average? Also in the same paragraph, you both use "coverage < 50%" and "converage in [10%, 30%]". Given the values used in the experiments, isn't this the same thing? I think, in general, the phrasing for results discussion could be improved to convey them more clearly.
	- In the abstract the acronym EBM is used but the term was not introduced yet.
	- Sec. 1, Par. 5: There is a question mark, possibly in place of a missing reference.
	- Sec. 1, Par. 6: Before (3) either it should be "challenge (i)" or there is a missing challenge, maybe the authors meant "challenges (i) and …".
	- Sec. 3.4, Par. 6: Referencing figure 3 when mentioning "step 1" and "step 2" would make it clearer to which steps the authors are referring to.
	- Sec. 5.1, Par. 7. The last phrase should be "provides up to", or "provides on average", regarding the reduction error of optimization based selection.
	- Sec. 5.3, Par. 1 and Figure 4: target coverage 100% is mentioned, however I do not see it in the figure. Is that a typo? I imagine target coverage 100% to be meaningless in this setting.
	- Sec. 5.1, Par. 7 and Sec. 5.4 seem to make conflicting statements about which energy inference method performs best.
	- A.1.1: Here it is stated that the energy optimization method achieves better prediction error of 1.9%, on average, over the aggregation method, while in Sec. 5.4, 10.9% is reported. I imagine this to be a typo, however, many average percentages are reported throughout the paper, focusing on different groupings of coverages and models. This might be confusing and I encourage the authors to find wordings that make it clear on which groupings of models and datasets these averages are computed.

**Strengths And Weaknesses:**

**Strenghts**
- The proposed methodology is novel and tries to address shortcomings of current EBM inference strategies with approaches that reduce the computational burden of sampling energy values in the context of a large output space like that of multi-step ahead time series forecasting.
- The provided results show that the approach is suitable as a plug-in component for different state-of-the-art time series forecasting models. Moreover, it compares favorably with respect to SelectiveNet which, as per authors claim, it is the only existing selective prediction approach suitable for adaptation to time series forecasting settings.

**Weaknesses**
- Details on how the different models are trained are not complete and, in absence of code for reproducibility, it is difficult to ensure that the bad performance of SelectiveNet is not due to implementation choices.
- The proposed approach is proposed in the context of univariate forecasting, i.e., the target variable is univariate. However, there is no discussion on the suitability of this approach for multivariate forecasting, where the target variable is multivariate. This is a common setting and it is not clear which is the applicability of the proposed approach for this scenarios in terms of performance and computational complexity. This discussion would be very useful for readers as parts of model training and inference involve iterative optimization or repeated sampling Eqs. (11, 13, 14).
- The experimental results mostly involve transformer based architectures, however architectures based on temporal convolutions networks and recurrent neural networks are still relevant approaches in time series forecasting. It would be nice to have an idea of the performance of the proposed approach when applied to such architectures.

**Questions**
- How is too low real coverage penalized when comparing performance of two selection models? Intuitively having an higher real coverage than the target one is not problematic and it is eventually implicitly penalized if the higher coverage is used to let bad predictions slip through. However, having a lower coverage should be penalized in some way as, in the extreme case, a selection method could result in a 0% coverage which technically results in a selective prediction error of 0, if my understanding is correct.
- Could the bad performance of SelectiveNet stem from the fact that its loss function results in a worse overall predictor? It is not clear if SelectiveNet is added on top of a pre-trained predictor, however, from my understanding, I imagine that SelectiveNet is trained from scratch using a weighted loss comprising an MSE term and a term related to the coverage. In this case the method could be very sensitive to the relative weight of the different components. Could the authors report the performance of the SelectiveNet predictor when the selection output is ignored and all predictions are considered ok? How does it compare to the performance of the corresponding standard supervisedly trained point-predictor?

---

> ### Author Response · Authors · 2025-01-04
> **Author response and clarifications**
>
> Thank you for your review and comments. We have addressed your comments and requested changes in the revised paper.
>
> - For question 1, we have added a discussion on how too low real coverage is penalized in Section 5.1, paragraph 3.
> - For question 2, we have provided details together with requested change 4 (see below).
>
> - For requested change 1, we have added more details on the models implementation and training in Appendix Section A.6.
> - For requested change 2, we have added a discussion on the suitability of the approach for multivariate forecasting in discussion Section A.11.
> - For requested change 3, we would like to note, that the TimesNet architecture is not transformer-based, but rather a convolutional architecture. We have added additional experiments with another, significantly smaller (in terms of parameter count) non-transformer based deterministic model -- LSTM in Section A.8. We have also added additional experiments with a probabilistic, non-transformer model TimeGrad in Section A.10.
> - For requested change 4, we have conducted additional experiments to evaluate the performance of SelectiveNet when trained using a traditional loss function (in this case, Mean Squared Error) and only using the prediction head. When trained this way, the SelectiveNet models showed the same deterministic forecasting performance as the base models. We have noted this in the revised paper, Section A.2.2. We have also provided more implementation details of how SelectiveNet models were adapted for forecasting in Section A.6.
> - For requested change 5, we have added the standard deviations across the 3 runs for each main TEM experiment in Appendix Section A.7.
> - For requested change 6, we have changed the results tables to increase readibility (see Tables 1,2,4). We have also clarified the meaning of the bold results in Section 5.1, paragraph 3.
> - For requested change 7, we have made corrections to the paper. We sincerely thank the reviewer for pointing them out.
>
> If there are any further changes or clarifications needed, please let us know.
>
> Thank you once again for your comments and for your time!

---

> ### Comment · Reviewer_3U5f · 2025-01-07
>
> I thank the authors for answering my comments. Following are some follow-up comments:
>
> RC1) From my understanding no hyper parameter selection was carried out, but values from the original papers were used. This also involves the lambda parameter of SelectiveNet, controlling the importance of coverage. Is this correct? I still find the implementation details possibly ambiguous, is there any reason preventing the release of code for reproducibility?
>
> RC4) There might have been a misunderstanding, I was interested in knowing the performance of SelectiveNet models, trained with the selection mechanism active, but while ignoring it at test time. Looking at the original SelectiveNet paper, this should roughly be equivalent to taking the output of the auxiliary head. However this is not critical, but it could be interesting to gather insight on whether the loss constraints in SelectiveNet harms the overall performance and if this could be mitigated by hyper parameter tuning.
>
> RC5) I think standard deviation should be reported for all models, not just TEM, and, possibly, for actual coverage too. Moreover, for readability, it should be reported within the respective tables rather than separately in the Appendix.
>
> RC6) I still find the main tables difficult to read, as models to be compared might be several lines apart. Is there a reason against sorting lines by "base model - coverage - selection method"? In this way the interesting comparisons would be easier to make at a glance.

---

> ### Author Response · Authors · 2025-01-18
> **Author response and clarifications**
>
> We sincerely thank the reviewer for their continued comments and suggestions.
>
> - For RC1:
> We have performed limited hyperparameter and architecture search for SelectiveNet models. This included experimenting with different architectures by attaching SelectiveNet heads to different parts of the base models (encoder and decoder), as well as learning rate adjustments. For a subset of base models (TimesNet, Informer, FEDformer), we also conducted experiments with different lambda parameters controlling the importance of coverage in SelectiveNet's loss function. However, these experiments did not yield performance gains beyond our current configuration. For the main experiments, we used the hyperparameters specified in the original papers for the base models (as detailed in Section 5.1 and Appendix A.6). The code will be made available for the camera-ready version.
>
> - For RC4:
> We thank the reviewer for the clarification. We have also conducted additional experiments with SelectiveNet models on a subset of base models (TimesNet, Informer, FEDformer), evaluating the performance while using SelectiveNet's loss function but only considering the auxiliary head output. These experiments showed that SelectiveNet's loss constraints do impact overall performance - the models consistently showed significantly higher prediction error compared to standard supervised training (as already reported in the paper). As mentioned in the answer before, we were unable to mitigate this performance gap through hyperparameter tuning.
>
> - For RC5:
> We have updated Table 1 with to include standard deviations for selective risk of all models.
>
> - For RC6:
> Thank you for the suggestion. We have reorganized Table 1 to group results by base model, then by target coverage, and finally by selective forecasting method (TEM or SelectiveNet). We hope this new organization makes it easier to directly compare the performance of different selective forecasting approaches for each model and coverage combination. We will apply the same reorganization to all results tables in the camera-ready version to maintain consistency throughout the paper, if the reviewer agrees.
>
> We thank the reviewer for their suggestions. If there are any further changes or clarifications needed, please let us know.

---

### Decision · Action_Editor_ZmAf · 2025-01-28

**Recommendation:** Accept as is

**Comment:**

The paper presents a time-series forecasting framework based on energy-based models, capable of estimating prediction error and rejecting predictions using energy functions. While one reviewer remains negative after the discussion, the majority of reviewers, along with the Action Editor (AE), agree that the claims in the submission are well-supported by clear evidence. The AE believes that researchers will find this work interesting, and practitioners will see value in applying this approach to real-world scenarios. Therefore, the AE recommends accepting the paper.

**Audience:**

Yes.

**Claims And Evidence:**

Yes.